ARTICLES

# Identification and structure of an extracellular contractile injection system from the marine bacterium *Algoriphagus machipongonensis*

Jingwei Xu [1,4], Charles F. Ericson[1,4], Yun-Wei Lien [1], Florentine U. N. Rutaganira [2], Fabian Eisenstein [1,3], Miki Feldmüller[1], Nicole King[2] and Martin Pilhofer [1]✉

Contractile injection systems (CISs) are phage tail-like nanomachines, mediating bacterial cell–cell interactions as either type VI secretion systems (T6SSs) or extracellular CISs (eCISs). Bioinformatic studies uncovered a phylogenetic group of hundreds of putative CIS gene clusters that are highly diverse and widespread; however, only four systems have been characterized. Here we studied a putative CIS gene cluster in the marine bacterium *Algoriphagus machipongonensis*. Using an integrative approach, we show that the system is compatible with an eCIS mode of action. Our cryo-electron microscopy structure revealed several features that differ from those seen in other CISs: a 'cap adaptor' located at the distal end, a 'plug' exposed to the tube lumen, and a 'cage' formed by massive extensions of the baseplate. These elements are conserved in other CISs, and our genetic tools identified that they are required for assembly, cargo loading and function. Furthermore, our atomic model highlights specific evolutionary hotspots and will serve as a framework for understanding and re—engineering CISs.

In most ecological settings, bacteria do not exist as isolated cells, but interact with other organisms[1]. These cell–cell interactions are often mediated by macromolecular machines that translocate effector proteins into the medium or directly into a target cell[2–4]. Bacterial contractile injection systems (CISs) mediate cell–cell interactions between bacterial and eukaryotic cells and often confer competitive advantage in different environmental niches. CISs are macromolecular injection devices with an overall structure that is homologous to the contractile tails of bacteriophages[5–8]. Their conserved modules include an inner tube, a contractile sheath and a baseplate complex. For firing of the extended apparatus, the baseplate undergoes a conformational change and triggers sheath contraction, which in turn causes the inner tube to be expelled and injected into a target.

On the basis of distinct modes of action, bacterial CISs are classified into extracellular CISs (eCISs) and type VI secretion systems (T6SSs). eCISs resemble headless phage particles that are assembled in the bacterial cytoplasm and then released into the medium upon cell lysis. Upon binding to a target cell via tail fibres[9], eCISs contract and puncture the target's cell envelope[10]. By contrast, T6SSs remain intracellular and are anchored to the inner membrane[11–14], injecting effectors by a cell–cell contact-dependent mechanism[15].

Classical T6SSs (subtypes i–iii) and R-type pyocins (eCISs) form relatively homogeneous groups of CISs. Recent bioinformatic analyses revealed an additional phylogenetic group of CISs with high abundance and diversity[16–19]. This group comprises hundreds of putative CIS gene clusters, with only a few of them being studied so far. A comprehensive bioinformatic study suggested that these CISs cluster into six distinct phylogenetic clades (Ia, Ib, and IIa–IId)[17]. Characterized representatives are only found in clades Ia and Ib and include 'Photorhabdus virulence cassettes' (PVCs)[20], 'antifeeding

prophages' (AFPs) from *Serratia*[21], 'metamorphosis-associated contractile structures' (MACs) from *Pseudoalteromonas luteoviolacea*[22], and the 'T6SS subtype iv' (T6SSiv) in 'Candidatus Amoebophilus asiaticus'[16]. While PVCs and AFPs in clade Ia act as individual, pyocin-like eCISs with insecticidal functions, MACs and T6SSiv in clade Ib were shown to have diverse functions. MACs form sea mine-like arrays of ~100 tethered eCISs that induce metamorphosis of larvae of the marine tubeworm *Hydroides elegans*, and additionally kill insect cells as well as murine macrophages in vitro[22,23]. The T6SSiv probably mediates the escape of symbiotic Amoebophilus bacteria from the phagosomes of its amoeba host[16].

Given the co-existence of systems with different modes of action and diverse targets in clade Ib, an atomic model of such a CIS assembly would be particularly insightful. The complicated superstructures in MACs and T6SSiv, however, impede their structural characterization[16,22]. Current high-resolution structural information is limited to clade Ia, that is, PVC and AFP[24,25].

Here we performed structural and mechanistic studies on a clade Ib CIS gene cluster in the marine bacterium *Algoriphagus machipongonensis* PR1. This strain was previously co-isolated with the choanoflagellate *Salpingoeca rosetta*[26] and was shown to induce the formation of multicellular colonies (rosettes) of *S. rosetta*[27–29]. Bioinformatic analyses showed that ~69% of bacteria in the *Algoriphagus* genus harbour a putative CIS gene cluster[19].

## Results and Discussion

**A gene cluster in *A. machipongonensis* encodes an eCIS.** We set out to characterize a putative CIS gene cluster in *A. machipongonensis* (hereafter referred to as AlgoCIS) that comprised 18 predicted open reading frames (accessions ALPR1_12680-12765/A3HTA7-A3HTC4). Within these open reading frames, we found sequence

¹Department of Biology, Institute of Molecular Biology & Biophysics, Eidgenössische Technische Hochschule Zürich, Otto-Stern-Weg 5, Zürich, Switzerland. ²Howard Hughes Medical Institute and Department of Molecular and Cell Biology, University of California, Berkeley, CA, USA. ³Present address: Graduate School of Medicine, University of Tokyo, N415, 7-3-1 Hongo, Bunkyo-ku, Tokyo, Japan. ⁴These authors contributed equally: Jingwei Xu, Charles F. Ericson. ✉e-mail: pilhofer@biol.ethz.ch

similarities with potential CIS structural components, besides additional proteins with unknown functions. Guided by sequence similarities to other CISs, we re-labelled the genes as *alg1-18* according to the homologues in AFP (Fig. 1a and Supplementary Table 1). Since there is no homologous protein of ALPR1_12705, a putative tail fibre with Ig-like folds, we labelled it *alg19*. Previous genome analyses classified CISs into six subtypes[17], with AlgoCIS being classified as Ib along with MACs[22] and T6SS[iv] (ref. [16]). This was further supported by our phylogenetic analyses (Fig. 1b).

To explore whether *A. machipongonensis* expresses any AlgoCIS particles, crude sheath purifications were performed and imaged by negative-stain transmission electron microscopy (EM). Typical CIS-like particles were found in both extended and contracted states (Extended Data Fig. 1a). Subsequent mass spectrometry (MS) analyses of the same sample detected 14 proteins encoded in the AlgoCIS gene cluster (asterisks in Fig. 1a and Supplementary Table 2). To further confirm these results, we disrupted the AlgoCIS gene cluster by inserting a plasmid into the AlgoCIS operon (AlgoCIS⁻). Since no CIS-like particles were detected in AlgoCIS⁻ mutant (Extended Data Fig. 1b and Supplementary Table 2), we conclude that the AlgoCIS gene cluster encodes CIS-like complexes.

To observe AlgoCIS in a cellular context, we imaged *A. machipongonensis* cells by cryo-electron tomography (cryoET). We frequently observed AlgoCIS particles in the bacterial cytoplasm but never anchored to the membrane (Fig. 1c), which is incompatible with a T6SS mode of action. CryoET analyses and western blots of bacteria at different optical densities ($OD_{600}$) indicated that the average expression level was highest (~3.8 AlgoCISs per cell) at high $OD_{600}$ growth phases (Extended Data Fig. 1c,d).

We further analysed the supernatants from wild-type and AlgoCIS⁻ cultures by western blot, negative-stain EM and MS. Using antibodies against Alg1 (inner tube) and Alg2 (sheath), we detected AlgoCIS in the supernatant of a wild-type culture (Extended Data Fig. 1e). We could also observe AlgoCIS particles by negative-stain EM (Extended Data Fig. 1f) and detected most putative structural components in the supernatant via MS (Supplementary Table 2). Taken together, these results indicate that AlgoCIS is consistent with an eCIS mode of action.

**Overall structure and unique features of the AlgoCIS particle.** To gain insights into the general structure of AlgoCIS, purified particles were imaged by cryoET. Sub-tomogram averaging showed that the AlgoCIS particle is ~130 nm long and ~30 nm wide, with the tube lumen filled with densities (Extended Data Fig. 1g,h). The structure can be divided into three modules: cap, sheath tube and baseplate. All three modules have 6-fold symmetric features (Extended Data Fig. 1i).

To reveal molecular high-resolution details, we imaged purified AlgoCISs for cryoEM single particle analysis and processed them as was previously shown for other CISs[24,25,30] (Extended Data Fig. 2a). The quality of the final maps from the three modules allowed for de novo structural modelling (Extended Data Fig. 3a,b, and Supplementary Fig. 1a and Table 3). Symmetry-related protein subunits were generated and merged into a complete model on the basis of the entire AlgoCIS map.

Out of the 18 proteins in the AlgoCIS gene cluster, 13 were localized in the final reconstruction, with 11 proteins having atomic models built (Fig. 1d). Two proteins with unknown functions (ALPR1_12695 and ALPR1_12690) were detected by MS, but were not identified in the cryoEM map (Supplementary Table 2).

The final model of AlgoCIS comprised 318 polypeptide chains, including 23 sheath layers (Alg2-L$_{1-23}$) and 22 inner tube layers (Alg1-L$_{1-22}$) (Fig. 1d). While our structural analyses showed a general overall agreement with reported structures of other eCISs[24,25], AlgoCIS represents the subtype Ib clade and revealed remarkable features (Fig. 1e).

**Cap and cap adaptor proteins terminate AlgoCIS.** The distal AlgoCIS end is terminated by a hexameric cap complex of Alg16A, and – in contrast to other known eCISs – a hexameric complex of Alg16B, hereafter referred to as the 'cap adaptor' (Fig. 2a). The dome-shaped Alg16A cap complex covers the inner tube, and it exhibits an ~11-Å-wide central channel (Fig. 2a and Extended Data Fig. 4a). Each Alg16A subunit interacts with three Alg1 proteins from the distal inner tube layer (Alg1-L$_{22}$) and terminates the tube (Extended Data Fig. 4b). Compared with homologues in PVC/AFP, Alg16A is much shorter (197 amino acids in Alg16A, 293 in Pvc16, 295 in Afp16) and folds into a single domain with a C-terminal extension. Structural superposition showed that the Alg16A structure is similar to the N-terminal domains (NTDs) of Pvc16 and Afp16 (Extended Data Fig. 4c).

Interestingly, the cap adaptor Alg16B mediates the interactions between the cap complex Alg16A and the most distal sheath layer (Alg2-L$_{23}$). The cap adaptor expands the outer diameter of the cap to ~164 Å, with an enlarged inner diameter of ~115 Å (Fig. 2a). The Alg16B NTD (residues 1–213) is an immunoglobulin-like domain flanked by three additional α-helices (α1–3), whereas one single β-barrel constitutes a C-terminal domain (CTD) (residues 214–284) (Fig. 2b). The Alg16B adopts a parallel arrangement with Alg16A (Extended Data Fig. 4d). The N terminus of Alg16B interacts with the C terminus of Alg16A in a mimicked 'handshake' manner, terminating the sheath assembly (Fig. 2c).

Next we set out to explore the role of the cap adaptor. Due to the lack of available precise genetic tools in *A. machipongonensis*, we developed a genetic toolset[31,32] to generate clean in-frame deletions (see Methods and Extended Data Fig. 4e,f). We thus created an AlgoCIS ΔAlg16B deletion and showed by cryoET imaging that it assembled different classes of aberrant AlgoCIS particles (Fig. 2d). Of the particles, ~49.8% ($n_{total} = 223$) had an overall similar shape compared with the wild type; however, sub-tomogram averaging showed that the cap, cap adaptor and the distal sheath layer (Alg2-L$_{23}$) were all absent (Fig. 2e). Another ~36.8% of particles contained only the inner tube and the baseplate, which is similar to the previously reported tube–baseplate complexes (TBC)[33]. Additionally, ~13.4% were TBC complexes with contracted sheath, similar to the 'contracted with jammed tube' particles seen in MACs[22]. Moreover, the tube lumen was empty for all observed particles in AlgoCIS ΔAlg16B (Fig. 2d and Supplementary Table 2). Together, these results showed that the cap adaptor plays an important role in stabilizing the cap module and the most distal sheath layer, and its absence results in mis-assembled particles.

**The sheath-tube module reveals diverse structural conformations across sheath layers.** Six protofilaments of sheath proteins adopt a right-handed helical array with a length of 92.4 nm and an outer diameter of 24.6 nm (Fig. 1d). Unlike the multiple sheath proteins present in PVC/AFP, the AlgoCIS sheath is composed of only one protein, Alg2, which folds into four domains (Extended Data Fig. 5a,b). The conserved domains I and II contribute to the sheath wall, with additional domains III and IV extending outwards (Extended Data Fig. 5c). The sheath subunits are interwoven with each other via a conserved 'handshake' and iteratively assemble into the full sheath, where the attachment helix in domain I mediates the interactions with the inner tube (Extended Data Fig. 5d,e).

Interestingly, depending on its location in the different sheath layers, Alg2 exhibits pronounced structural variations. Both the N and C termini of Alg2 have different conformations in the proximal (Alg2-L$_1$), distal (Alg2-L$_{23}$) and central sheath layers (Alg2-L$_{2-22}$) (Extended Data Fig. 5f). The hinge angles of domain IV were also observed to be different when comparing the different Alg2 layers. The domain IV in the distal Alg2-L$_{23}$ layer has the smallest hinge angle, with its tip close to domain II from the neighbouring subunit in the same sheath layer (Extended Data Fig. 5f,g). The different

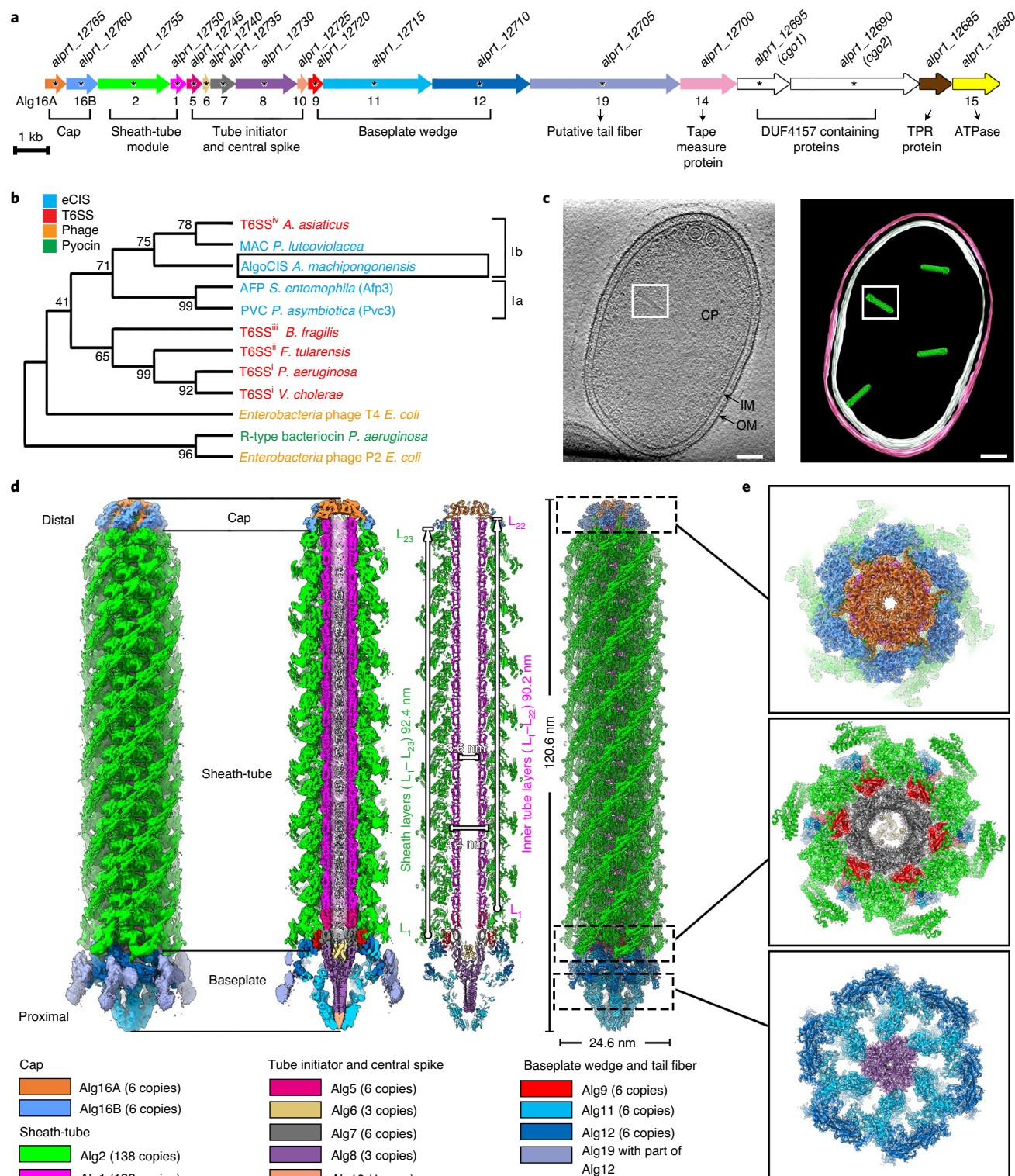

**Fig. 1 | Identification and characterization of a contractile injection system in *A. machipongonensis*. a**, Schematic showing the gene cluster of a putative contractile injection system in *A. machipongonensis* (AlgoCIS). The genes are labelled on the basis of similarities to AFP. Gene products that were detected by MS are marked by asterisks. The gene accession numbers are shown above the corresponding genes. **b**, Phylogenetic analyses based on putative sheath proteins showing that the closest relatives of AlgoCIS are MACs and T6SS[iv], which belong to clade Ib CIS. The different representatives are colour-coded on the basis of their modes of action. **c**, Representative cryoET slice of *A. machipongonensis* cell (left) and the corresponding model (right), showing cytoplasmic AlgoCISs that are not attached to the inner membrane. CP, bacterial cytoplasm; IM/white, inner membrane; OM/pink, outer membrane; green, AlgoCIS particle. One representative AlgoCIS particle is marked with a white box. Shown is a 10.8 nm thick slice. Scale bars, 50 nm. In total, 38 tomograms were acquired. **d**, Shadowed surface (left) and ribbon (right) diagrams showing the overall cryoEM structure of AlgoCIS in the extended state (sliced views in the centre). Structural subunits are colour-coded according to the gene cluster in **a**. **e**, Perpendicular views of shadowed surface and ribbon diagrams showing the AlgoCIS model corresponding to the sections in **d**.

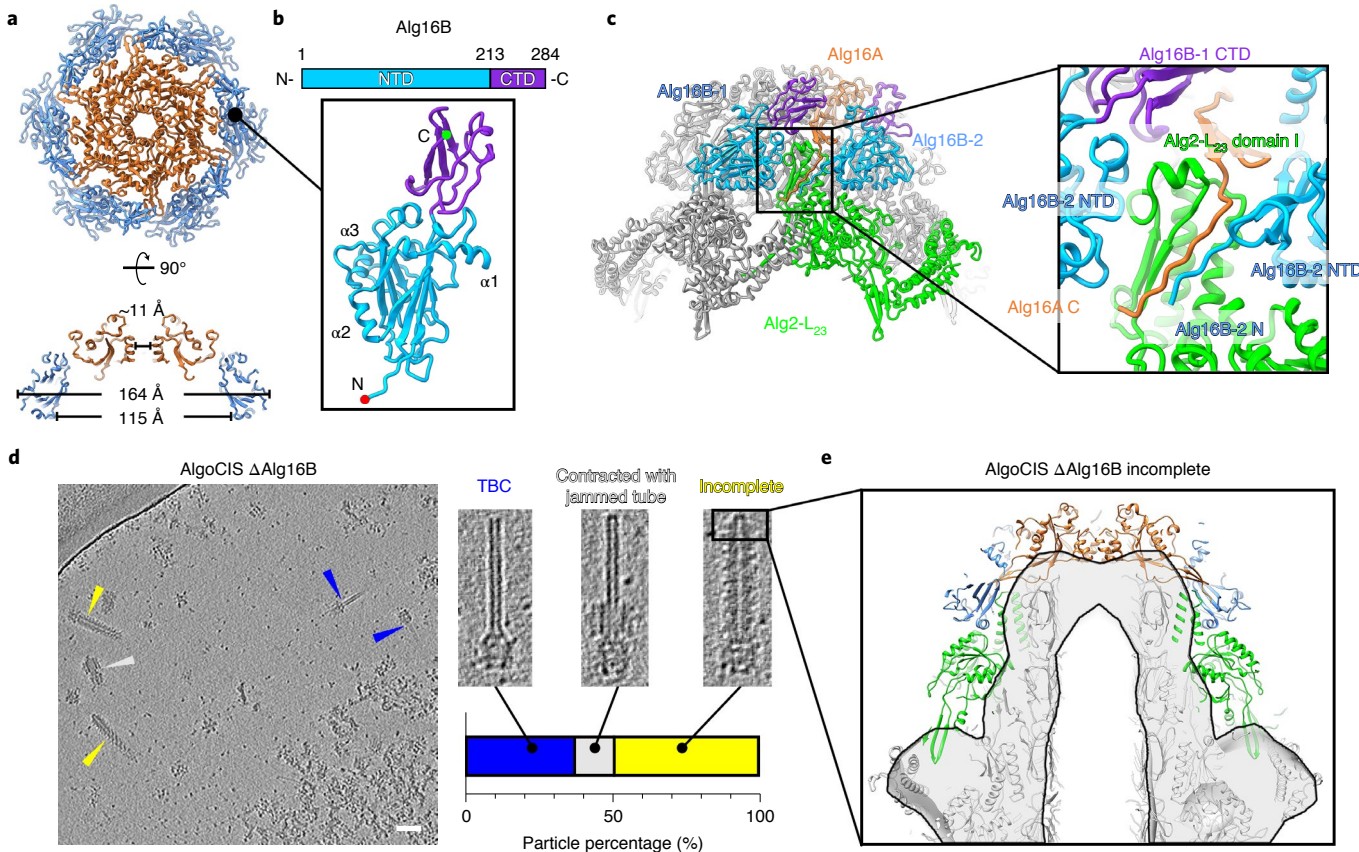

**Fig. 2 | Cap and cap adaptor proteins terminate AlgoCIS. a**, Top view (top) and side view (bottom) of ribbon diagrams showing that cap proteins (Alg16A) interact with cap adaptor proteins (Alg16B) and together assemble into the cap module. The colour code matches Fig. 1d. **b**, Schematic and ribbon diagrams showing that the cap adaptor protein (Alg16B) has two domains. The N- and C-terminal domains (NTD and CTD) are cyan and purple, while the N and C termini are marked with red and green circles. **c**, Left: ribbon diagrams of cap module showing that Alg16A and Alg16B adopt a mimicked hand-shaking manner to terminate the sheath-tube module. The colour code for one individual protomer of Alg16A and distal sheath layer (Alg2-$L_{23}$) matches Fig. 1d, whereas two protomers of cap adaptor protein (Alg16B-1 and Alg16B-2) are coloured on the basis of the domain organization in **b**. The box indicates the interactions between cap module and sheath protein. Right: the structural detail of the box on the left. **d**, CryoET of AlgoCIS ΔAlg16B particles reveals three different classes of aberrant particles. Shown is an overview image, three examples for the different classes, and a quantification (11 nm thick tomographic slices). Scale bar, 50 nm. **e**, Sub-tomogram averaging of the AlgoCIS ΔAlg16B 'incomplete' class from **d**, revealing that the cap, cap adaptor and the distal sheath layer are absent. Shown is a comparison of the wild-type model (ribbon diagram) and the average (grey and black). The proteins absent in AlgoCIS ΔAlg16B are colour-coded according to Fig. 1d.

conformations of N and C termini in the most proximal (Alg2-$L_1$) and most distal sheath layers (Alg2-$L_{23}$) might mediate the assembly of the sheath-tube module. A lowpass-filtered map indicates that there are two potential conformations of domain IV in the central sheath layers (Extended Data Fig. 5h).

Regarding the inner tube, the 22 hexameric layers (Alg1-$L_{1-22}$) arrange in the same helical parameters as the extended sheath, forming a 90.2-nm-long conduit with a 3.6-nm-wide tube lumen (Fig. 1d). The inner tube subunits possess the conserved β-barrel structures and the N terminus of Alg1 is shorter than that of Pvc1 and Afp1 (Extended Data Fig. 5i). The inner surface of the tube is negatively charged as in other CISs[24,25,34] (Extended Data Fig. 5j).

**The structures of the tube initiator and central spike reveal unique domain organization.** The inner tube is attached to the central spike by the tube initiator complex (Alg5/Alg7), which is further docked onto the spike (Alg6/Alg8/Alg10) (Fig. 3a). All components of the tube initiator complex possess the conserved β-barrel folds seen in Alg1 (Extended Data Fig. 6a), whereas Alg7 has one additional C-terminal LysM domain extending out and interacting with the peripheral wedges (Fig. 3a and Extended Data Fig. 6a).

Surprisingly, the linker between the β-barrel and LysM domain in Alg7 was found to be cleaved (Extended Data Fig. 6b).

The spike is located below the tube initiator complex and comprises three intertwined copies of Alg8 (Fig. 3a). The N-terminal part of Alg8 forms a gp27-like domain and functions as a 3-to-6-fold symmetry adaptor[35]. The gp27-like domain is followed by the mid β-barrel domain and C-terminal gp5-C-like domain, which are homologous to T4 gp5 (Fig. 3b). Intriguingly, the gp5-C-like and mid β-barrel domains in Alg8 twist an additional 120° around the central axis and are much longer when superposing the gp27-like domains against homologues in PVC/AFP (Fig. 3c). The tip of the central spike binds to a single copy of Alg10, which is a proline–alanine–alanine–arginine (PAAR)-like protein (Fig. 3a).

**The Alg6 plug protein is crucial for CIS assembly and function.** Interestingly, our symmetry-relaxed map showed an additional prominent density plugged into the tube-exposed cavity of the spike. We identified this density by structural modelling unambiguously as a trimer of the uncharacterized protein Alg6 (Fig. 3d and Extended Data Fig. 6c), hereafter referred to as the 'plug'. Alg6 is an orthologue of Afp6, which was previously not resolved[25]. Alg6 includes an

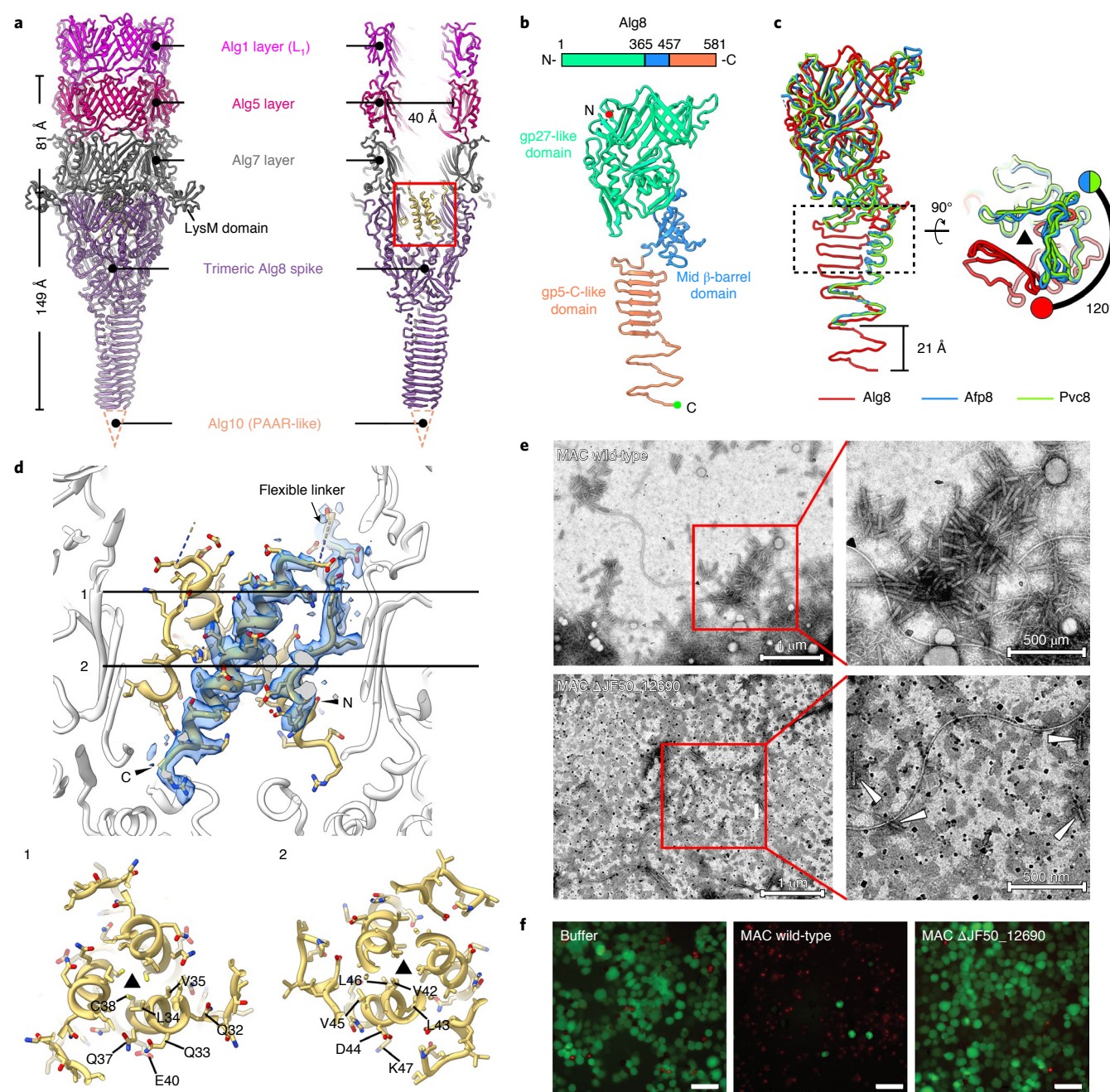

**Fig. 3 | The Alg6 plug protein is crucial for assembly and function. a**, Side (left) and cutaway (right) views of ribbon diagrams showing the overall structure of the central part of the AlgoCIS baseplate. The colour code matches Fig. 1d, while the PAAR-like Alg10 is represented by the dashed triangle. The box indicates the position of trimeric plug protein Alg6, which is shown in **d**. **b**, Schematic and ribbon diagrams showing that Alg8 has three domains. The N and C termini are marked with red and green circles. **c**, Left: structural superpositions of Alg8 (red) and homologues (Afp8, blue; Pvc8, green) based on the gp27-like domain showing the different domain organization in the Alg8 protein compared with the homologues. Right: perpendicular view of the dashed box on the left. The black triangle indicates the symmetry axis. **d**, Top: shadowed surface and ribbon diagrams showing the plug (trimeric Alg6) localization inside the tube-exposed cavity of the central spike. The corresponding densities surround one Alg6 protein are coloured transparent blue. The positions of the flexible linker, N and C termini are labelled. Bottom: ribbon diagrams of perpendicular slices (positions 1 and 2 in the top panel), revealing that the hydrophobic core of the Alg6 trimer exposes charged residues on the outer surface. **e**, Negative-stain EM images of purified MACs from wild-type *P. luteoviolacea* and mutant ΔJF50_12690, showing aberrant MAC tail structures (highlighted by white arrowheads) in the strain with deleted Alg6 homologue (JF50_12690). This experiment was performed three independent times. **f**, Killing assays showing that the deletion of the Alg6 homologue in MACs (JF50_12690) dramatically impairs the killing effect against Sf9 insect cells. Shown are fLM images of Sf9 cells. Red (propidium iodide), dead cells; green (fluorescence diacetate), live cells; buffer, control (no MACs added). Scale bars, 50 μm. This experiment was performed three independent times with biological replicates.

N-terminal single strand and a C-terminal α-helix, which are linked by a flexible loop (Extended Data Fig. 6c). The Alg6 N terminus extends along the inner surface of the Alg8 gp27-like domain, then folds back and forms the C-terminal α-helix (Fig. 3d). The hydrophobic core of Alg6 contributes to the trimeric assembly, directing the hydrophilic surface to interact with other baseplate components (Fig. 3d).

There are extensive contacts between Alg6, Alg8 and Alg7 (Extended Data Fig. 6d,e), suggesting that Alg6 is essential and functions as a nucleus for particle assembly. To test this hypothesis, we generated and analysed a ΔAlg6 mutant. Remarkably, no assembled AlgoCIS particles were found in a sheath preparation of the knockout (Extended Data Fig. 6f). We also detected putative 'plug' homologues in the closely related MACs, as well as in the T6SS[iv] (accessions JF50-12690/Aasi_1078) (Extended Data Fig. 6g). We deleted the plug homologue in MACs in *P. luteoviolacea* and purified mutant MACs[23,36]. Negative staining resulted in no detectable MAC arrays (Fig. 3e). Since wild-type MAC purifications were previously shown to kill insect cells[23,36], we tested MACs ΔJF50-12690 and found that they had a dramatically impaired ability to kill insect cells (Fig. 3f). Together, these data show that the small Alg6 plug – and its homologues in other CISs – are crucial for CIS assembly and function.

**Conserved baseplate components with downward extending tail fibres.** The central spike is surrounded by a hexagonal iris-like ring of Alg11-Alg12 heterodimers, forming the baseplate 'wedges' with the conserved core bundle and trifurcation units (Fig. 4a). Six copies of the gp25-like protein (Alg9) attach above Alg11-Alg12, where the C termini from two neighbouring protomers were observed to be slightly different (Extended Data Fig. 7a). Like gp25 in the T4 phage[37], Alg9 mimics the 'handshake' to initiate sheath assembly and it interacts extensively with other baseplate components, implying a role in the initiation of contraction (Fig. 4a and Extended Data Fig. 7b).

Similar to Pvc12 and Afp12, Alg12 contains gp6-like and gp7-like parts (Extended Data Fig. 7c). Domains I–III constitute the gp6-like part and participate in the assembly of the wedges, leaving the gp7-like part (domains IV–VII) exposed on the lateral surface of the baseplate (Fig. 4a). Three-dimensional-focused classification revealed diverse conformations for the gp7-like part of Alg12 (Extended Data Fig. 7d). It is responsible for interactions with tail fibres. In contrast to AFP/PVC[24,25], the AlgoCIS tail fibres do not fold back to contact the sheath, but instead attach on the lateral surface of the wedges and extend downwards (Fig. 1d). CryoEM 2D projection images showed that tail fibres adopt various conformations (Extended Data Fig. 7e).

**Alg11 extensions form a unique spike cage.** AlgoCIS exhibits a remarkable structural feature on the baseplate—the formation of an extensive 'cage' around the central spike. Therefore, Alg11 is much longer and folds into six domains compared with the homologues (Fig. 4b). The Alg11 domain I forms the conserved core bundle together with the domains I and IV in Alg12 (Fig. 4a). The Alg11 N terminus extends across the cleft between domain I and VI (Fig. 4b and Extended Data Fig. 7f). Interestingly, there is one additional large domain (IV, residues 278–643) protruding from the distal tip

of domain III (Fig. 4b). The domains IV from the six protomers form a hexagonal cage that surrounds the spike (spike cage). The outer surface of the spike cage is negatively charged, with one positively charged residue at the tip (Lys572) (Fig. 4c and Extended Data Fig. 7g). The lumen of the spike cage ranges in diameter from 56 Å to 18 Å and a tip cavity is filled by Alg10 (Fig. 4c). A Dali search[38] suggested that domain IV comprises two carbohydrate-binding modules (CBM1 and CBM2) (Fig. 4b and Extended Data Fig. 7h) and the key residues mediating sugar binding in the protein TmCBM27 (PDB entry: 1OF4)[39] are conserved in CBM1 of Alg11 (Extended Data Fig. 7h). Thus, we hypothesize that the spike cage might be involved in AlgoCIS attachment.

Since a spike cage has never been seen in any of the previous high-resolution structures, we investigated its presence in the closest AlgoCIS relatives, namely in MACs and T6SS[iv]. We analysed the sequences of Alg11 homologues and indeed found the characteristic extensions. Subsequent sub-tomogram averaging of both the MAC and T6SS[iv] baseplates revealed that both also contained cage-like structures that were similar to AlgoCIS (Fig. 4d). Thus, the spike cage complex seems to be conserved across different modes of action in representatives of eCISs and T6SS[iv].

**Conformational changes of the sheath and baseplate cage after contraction.** To explore the mechanism of AlgoCIS firing, we determined post-firing structures of both sheath (Extended Data Figs. 2b, 3c,d and Supplementary Fig. 1b) and baseplate (Supplementary Fig. 2). Similar to other CISs[24,25,40,41], the displacements of N and C termini of the sheath subunits result in a similar rigid-body rotation of the sheath subunits, leading to sheath contraction (Supplementary Fig. 3).

In the contracted state, the spike cage adopts an open conformation, with the inner tube expelled across its centre (Extended Data Fig. 8a). The conformational change of the spike cage is mainly attributed to the movements of domain IV of Alg11, which tilts an additional ~27.4° outwards (Extended Data Fig. 8b). In addition, the tail fibres show a large outward tilt (~57°) (Extended Data Fig. 8c). Strikingly, the overall structure of the tail fibre in the contracted state has a shape that is similar to the T6SS baseplate component TssK (PDB entry: 5MWN)[42] (Extended Data Fig. 8d).

Furthermore, upon contraction, the baseplate protein Alg12 has a higher hinge angle with the plane of the iris-like ring (~66° in contracted, ~48° in extended state) (Extended Data Fig. 8e). Interestingly, the iris-like ring structure in AlgoCIS was found intact and having a slight expansion upon contraction (Extended Data Fig. 8a), which is different from the contracted structures in both pyocins and AFP[25,30].

Given the conformational changes that were observed in the baseplate components, we speculate that AlgoCIS might employ a similar signal transmission mechanism as the T4 phage (Extended Data Fig. 8f): the large outward tilting of tail fibres causes the

**Fig. 4 | A cage surrounds the AlgoCIS spike and is also seen in other CISs. a**, Cutaway view of ribbon diagram showing the overall structure of the AlgoCIS baseplate and revealing the unique spike cage. The central part of the baseplate is coloured white, while the colour code for Alg1-L$_{23}$ and Alg9 matches Fig. 1d. The gp6-like and gp7-like parts of Alg12 are coloured yellow and cyan, respectively. The major parts of Alg11 are coloured purple, with the domain IV coloured cherry. The dashed magenta box indicates the core bundle in the AlgoCIS baseplate. The shadowed surface and ribbon diagrams of perpendicular slices of positions 1 and 2 are shown in the right panels. The baseplate map is lowpass filtered to 5 Å. The conserved trifurcation unit and the central spike are highlighted by a red dashed triangle and a black dashed circle, respectively. **b**, Schematic and ribbon diagrams showing that Alg11 has six domains. The N and C termini are marked with red and green circles. Two carbohydrate-binding modules (CBM1/2) and the tip loop are labelled. **c**, Left: cutaway view of ribbon diagram showing the novel spike cage structure in the AlgoCIS baseplate. Two carbohydrate-binding modules (CBM1/2) and the tip loop are labelled, and the side chain of the positively charged residue (Lys572) is shown at the tip of the cage. Right: bottom view of dashed box on the left panel showing that the PAAR-like protein Alg10 (orange) fills the tip lumen of the spike cage. **d**, Structural comparisons of the AlgoCIS high-resolution structure with sub-tomogram averages of purified MACs (left) and in situ T6SS[iv] (right). The top row shows shadowed surface diagrams (grey) of sub-tomogram averages. The bottom row shows structural dockings of the AlgoCIS baseplate into the averages (white contours), revealing the presence of a spike cage (outlined by dashed lines) in all three CISs. The colour code of different AlgoCIS components matches that in **a**. The asterisks (cyan) indicate the lateral regions of the gp7-like part of Alg12. The triangles (orange) indicate the additional tail-fibre-like protein binding sites in MACs and T6SS[iv].

rotation and outward motion of baseplate components; the signal is subsequently transferred to the Alg11/12 core bundles and contributes to the tilts of the core bundle and the bound proteins; the conformational changes of Alg9 further trigger sheath contraction.

**ALPR1_12695 (Cgo1) and ALPR1_12690 (Cgo2) are cargo proteins located in the tube lumen.** In addition to the CIS structural proteins, the AlgoCIS gene cluster encodes two DUF4157 domain-containing proteins with unknown functions (Fig. 1a).

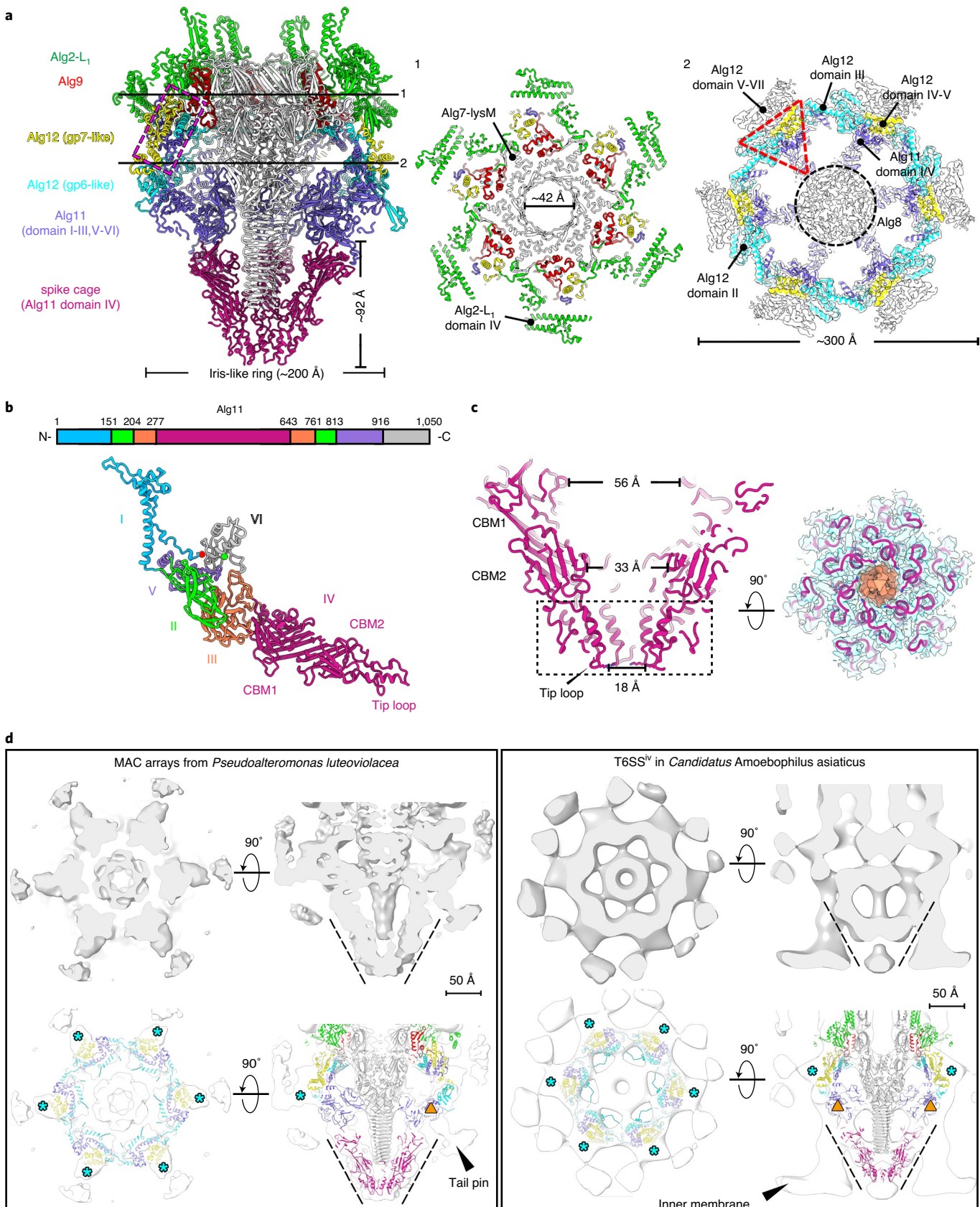

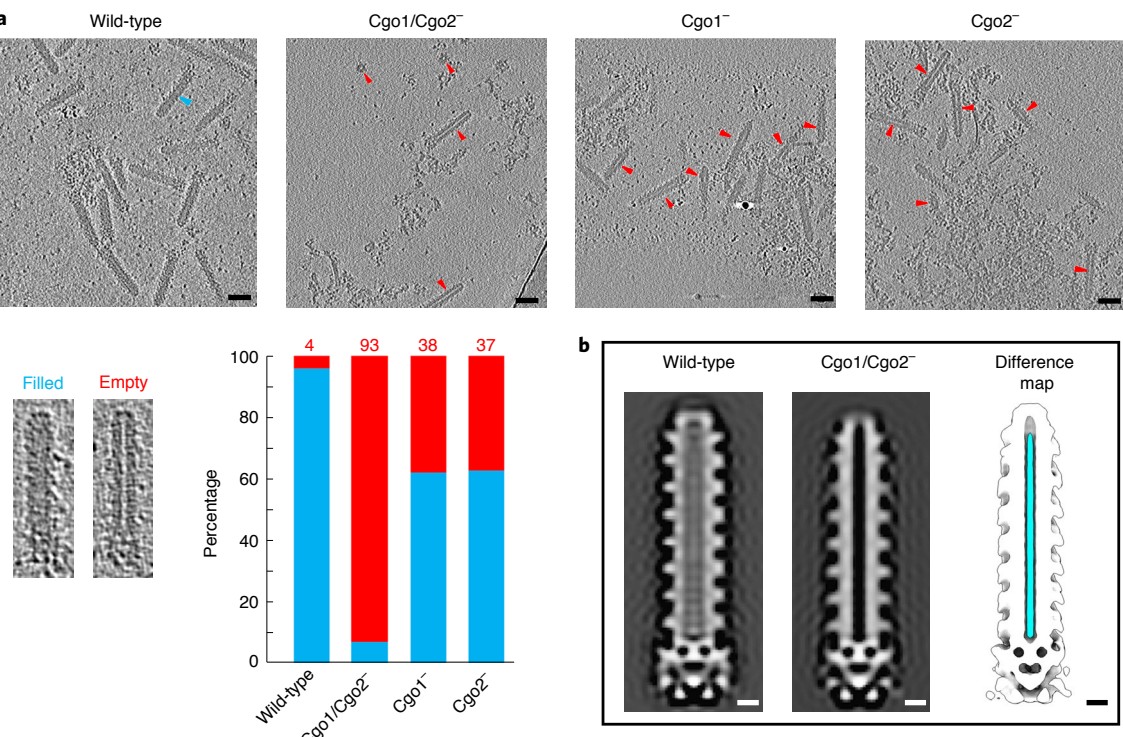

**Fig. 5 | Cgo1 (ALPR1_12695) and Cgo2 (ALPR1_12690) are cargo proteins filling the tube lumen. a**, Top: cryoET slices of purified wild-type AlgoCIS and different deficient mutants showing that the fractions of empty AlgoCIS particles are significantly increased in mutants (wild-type, $n_{total} = 2,620$; Cgo1/ Cgo2⁻, $n_{total} = 543$; Cgo1⁻, $n_{total} = 1,383$; Cgo2⁻, $n_{total} = 1,104$). Bottom left: representative filled and empty AlgoCIS particles. Bottom right: quantification (blue, filled; red, empty). The individual empty AlgoCIS particles are highlighted by red arrowheads, whereas one filled AlgoCIS particle is highlighted by a blue arrowhead. Tomographic slices are 10.8 nm thick. Scale bars, 50 nm. **b**, Central volume slices showing the sub-tomogram averages of wild-type AlgoCIS (left) and Cgo1/Cgo2⁻ (middle). The difference map (right) between wild-type and Cgo1/Cgo2⁻ mutant shows additional densities in cyan (corresponding to cargo proteins) and the remaining parts in white. Scale bars, 10 nm.

We will hereafter refer to these genes as *cgo1* and *cgo2*, respectively. Cgo1/Cgo2 share a similar predicted disordered NTD and a central DUF4157 domain, while their CTDs are different (Extended Data Fig. 9a). Interestingly, DUF4157 domains were previously identified as characteristic effector domains in eCISs and T6SSs[43]. Due to the closely related MACs having an effector/ cargo protein that was previously reported to localize to the tube lumen[36], we set out to determine whether Cgo1/Cgo2 share this cargo role in AlgoCIS.

To test this hypothesis, we generated mutants by abolishing the expression of Cgo1/Cgo2 individually, or both (Extended Data Fig. 9b). CryoET revealed that the wild-type inner tube was almost always filled (~96%, $n_{total} = 2,620$) (Fig. 5a and Extended Data Fig. 1j). In contrast, the inner tube was mostly empty (~93%, $n_{total} = 543$) in the Cgo1/Cgo2⁻ mutant and the occupancy was dramatically reduced in the individual Cgo1⁻ and Cgo2⁻ mutants (~38%, $n_{total} = 1,383$ and ~37%, $n_{total} = 1,104$, respectively) (Fig. 5a). The difference map between the sub-tomogram averaging volumes of wild-type and double mutant also exhibits a continuous density occupying the entire inner tube lumen (Fig. 5b).

Since cryoEM of contracted AlgoCIS always showed empty inner tubes (Extended Data Fig. 7e and Supplementary Fig. 2c,d), we hypothesized that the Cgo1/Cgo2 proteins leak from the inner tube after contraction, similar to the cargo in MACs[36]. To corroborate this idea, we compared the preparations of wild-type AlgoCIS in extended vs contracted states by MS. Consistent with our hypothesis, Cgo1 was missing in the MS data of contracted AlgoCIS samples, while Cgo2 was detected with a significantly smaller number of unique peptides (Supplementary Table 2).

Due to our observation that Cgo1/Cgo2 are located in the tube lumen, we explored the protein–protein interactions of the two cargo proteins and determined whether they could be co-loaded into the same tube. Our results indicated that the cargo proteins probably do not specifically interact with one another (Extended Data Fig. 9c). Nevertheless, we could not exclude the possibility that two cargo proteins are loaded into the same tube without strong contacts. Together, we conclude that Cgo1/Cgo2 are both cargo proteins that can be loaded independently from each other into the lumen of the inner tube, from which they are released after contraction.

**Heterologous expression of Cgo1 inhibits bacterial growth.** Sequence analyses predicted that the Cgo1 CTD might encode a metalloprotease (Extended Data Fig. 9a). To further explore a possible function of AlgoCIS, the Cgo1/Cgo2 were recombinantly expressed in *Escherichia coli*. The expression of the full-length Cgo1 protein inhibited growth of the bacteria and the effect of Cgo1 was independent of the metalloprotease motifs in DUF4157 and CTD domains. Furthermore, neither the fusion of a periplasmic translocation tag (Tat system), nor the co-expression of Cgo1/Cgo2 mitigated the inhibiting effect of Cgo1 (Extended Data Fig. 9d). To explore a possible anti-bacterial effect of AlgoCIS particles, we co-incubated them with different bacterial strains; however, none of the tested strains were sensitive to AlgoCIS (Extended Data Fig. 10a). This is not surprising, since eCISs such as pyocins can be highly specific to individual bacterial strains[44,45].

Besides many CISs having an anti-bacterial effect, some eCISs were reported to mediate interactions with eukaryotic cells[36,46,47]. We co-incubated a range of potential eukaryotic targets with

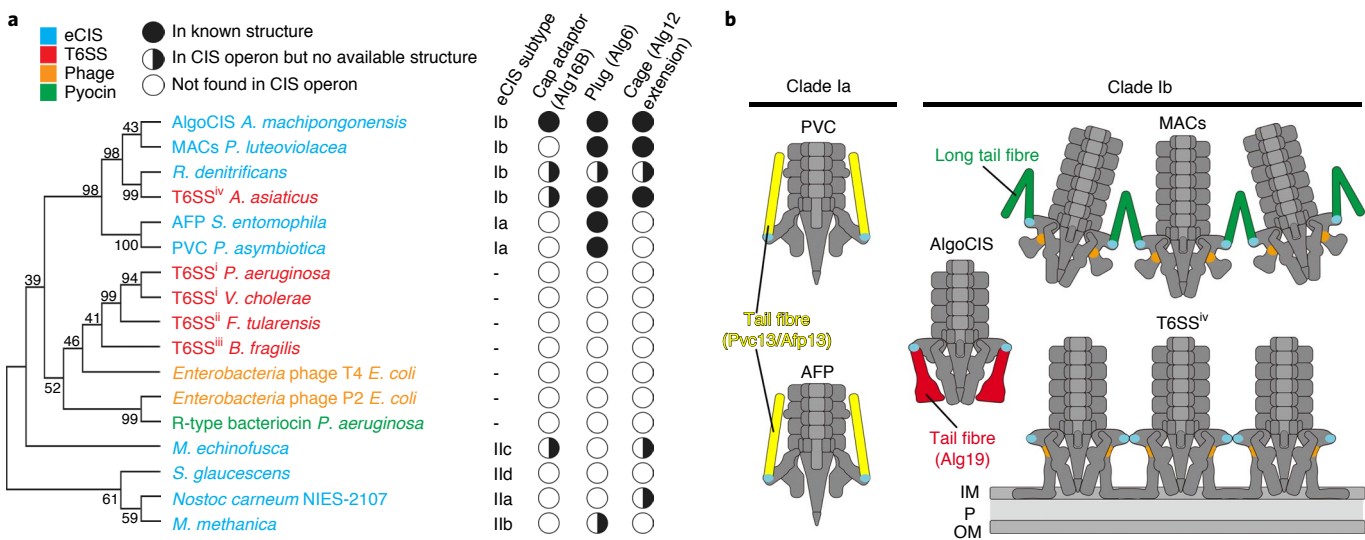

**Fig. 6 | Conservation of unique structures and hotspots for evolutionary re-engineering. a**, Phylogenetic tree based on phylogenetic analyses of putative sheath proteins. The columns on the right indicate the presence of homologues of the cap adaptor, plug and spike cage in different CISs. **b**, Schematic identifying the gp7-like part of Alg12 (cyan; also see Fig. 4d) as a hotspot for evolutionary re-engineering, giving rise to different superstructures and binding to different types of tail fibres. Furthermore, additional sites at the spike cage (orange; also see Fig. 4d) may be critical for the conversion between eCIS (MACs) and T6SSs (T6SS[iv]). IM, inner membrane; P, periplasm; OM, outer membrane.

*A. machipongonensis* bacteria. The comparison of assays performed with wild-type vs AlgoCIS⁻ mutant bacteria showed no significant differences (Extended Data Fig. 10b). The same was true when we tested the effect of purified AlgoCIS with insect cells—a known target of MACs[23] (Extended Data Fig. 10c). Since *A. machipongonensis* was reported as a potential bacterial prey of the choanoflagellate *S. rosetta*[48], we also hypothesized that AlgoCIS might affect the proliferation of the choanoflagellate. However, no significant effect was detected when the intact bacteria or purified AlgoCIS were incubated with the choanoflagellates (Extended Data Fig. 10b,d).

## Conclusions

Besides the classical T6SSs (subtypes i–iii) and pyocins (eCISs), recent bioinformatic genome analyses uncovered a significant additional phylogenetic group of CISs[16–18]. In addition to their abundance across sequenced bacterial genomes, an intriguing feature of this group is the co-existence of eCISs and T6SSs among very close relatives in clade Ib. The high-resolution structures presented here and in an accompanying study[49], will serve as a framework to understanding the evolution and function of these related systems. In this regard, our discovery of features such as the cap adaptor, plug and spike cage are significant, since we found that they were conserved in many other systems (Fig. 6a).

Another major insight is the identification of evolutionary hotspots. Our data suggest that the gp7-like part of the baseplate wedge-component Alg12 and its homologues mediate the formation of superstructures (Fig. 6b). In T6SS[iv], up to 34 individual T6SS[iv] structures form ordered arrays, mediated by lateral interactions of the baseplates[16]. In MACs, ~100 individual eCISs form ordered arrays, mediated by long tail fibres that connect neighbouring eCISs[22]. The structural comparisons enabled by the AlgoCIS structure presented here suggest that the gp7-like baseplate components are probably involved in mediating these interactions – directly in T6SS[iv] and by binding the long tail fibres in MACs (Fig. 6b, cyan). AlgoCISs, AFPs and PVCs are thought to act as individual eCISs and the gp7-like baseplate component (Fig. 6b, cyan) mediates contacts with different types of tail fibres.

The AlgoCIS tail fibres themselves reveal a surprising degree of compositional and structural heterogeneity among related

assemblies. No homologues of Alg19 were found in AFP/PVC or MACs, which feature different types of tail fibres that emanate from the baseplate and are oriented towards the distal end/sheath. Alg19, however, rather adopts an orientation downwards, with the general structure being found to be similar to the corresponding region in the canonical T6SS baseplate component TssK.

A second evolutionary hotspot that is revealed by the comparison of the AlgoCIS structure with sub-tomogram averages of MACs and T6SS[iv] is a binding site at the cage domain of Alg11 homologues (Fig. 6b, orange). This site mediates the attachment of short tail fibres in MACs and the attachment of the entire apparatus to the inner membrane in T6SS[iv]. Eventually, our data may enable structural and functional predictions for uncharacterized CIS gene clusters in the future.

Finally, our genetic tools for *A. machipongonensis* enable future approaches to explore the molecular details of AlgoCIS assembly, signal transmission and contraction in the natural host organism. The combination of the AlgoCIS structure, together with another concurrently reported system in cyanobacteria[49], the sub-tomogram averages of other clade Ib CISs, and the genetic tools will facilitate targeted approaches to re-engineer AlgoCIS for biomedical applications and may even allow for switching its mode of action.

## Methods

**Sheath preparation of AlgoCIS.** The AlgoCIS purification was followed as previously reported with some modifications[50]. Briefly, a small volume of *A. machipongonensis* was inoculated into 1 l fresh marine broth (MB) medium (Condalab) and grown at 30 °C and 200 r.p.m. for 2 d (bacterial OD₆₀₀ = ~5.0). The additional antibiotic erythromycin was added to the medium at the final concentration of 50 μg ml⁻¹ for culturing deficient mutant strains. The bacterial pellet was harvested by centrifugation and resuspended with buffer A (20 mM Tris pH 8.0, 150 mM NaCl, 50 mM EDTA). The lysis reagents (1% Triton-X100, 0.5× CellLytic B (Sigma-Aldrich), 200 μg ml⁻¹ lysozyme, 50 μg ml⁻¹ DNAse I, and protease inhibitor cocktail (Roche)) and 1.5 mM final concentration of MgSO₄ were added into the bacterial suspension and incubated at 37 °C for 30 min to lyse the bacteria. The cell debris were removed by centrifugation at 15,000 g and 4 °C for 20 min. The supernatant was subjected to ultra-centrifugation with sucrose cushion (1 ml at bottom) (20 mM Tris pH 8.0, 150 mM NaCl, 50 mM EDTA, 1% Triton-X100, 50% (w/v) sucrose) at 150,000 g and 4 °C for 1 h. The sucrose cushion was taken, together with some remaining overlying liquid (~ 0.5 ml). The residual bacterial contamination in solution was further removed by centrifugation at 15,000 g for 15 min. The sample was subjected to a second

round of ultra-centrifugation without a sucrose cushion. The resulting pellets were washed, soaked with buffer B (20 mM Tris pH 7.5 and 150 mM NaCl) overnight, and then resuspended. The crude samples were subjected to negative-stain EM imaging and mass spectrometry.

For cryoEM analysis, the crude sample was further purified through a 10–50% (w/v) sucrose gradient at 100,000 g and 4 °C for 1 h using the SW 55 Ti rotor. The gradient was divided into 11 fractions and each fraction was checked for AlgoCIS with negative staining EM. The fractions containing AlgoCIS were diluted with buffer B, passed through a 0.1-μm-pore filter twice, and then concentrated by a third round of ultra-centrifugation. The pellets were resuspended in buffer B and filtered through 0.22 μm centrifugal filtering (Millipore).

The contraction of AlgoCIS was performed following a previously reported method[25]. The purified AlgoCIS samples were incubated with 2 M guanidine-HCl at room temperature for 0.5 h. The guanidine-HCl was replaced by buffer B using Slide-A-Lyzer MINI dialysis devices (ThermoFisher). The contracted samples were subjected to cryoEM vitrified sample preparation. To separate the released cargo proteins from contracted AlgoCIS, the samples were ultra-centrifuged at 150,000 g and 4 °C for 1 h. The pellets were resuspended in buffer B.

**AlgoCIS purification from bacterial supernatant.** The purification of released AlgoCIS from the bacterial supernatant was performed using the $(NH_4)_2SO_4$ precipitation method[50]. Briefly, different *A. machipongonesis* strains (wild type, AlgoCIS⁻ and null mutant) were inoculated in 50 ml MB broth and incubated at 30 °C, shaking at 200 r.p.m. for 2 d. The bacterial cells were centrifuged at 7,000 g for 20 min and the supernatant was taken for subsequent purification. The saturated $(NH_4)_2SO_4$ solution was slowly added into the supernatant to a final concentration of 1.4 M and kept stirred at 4 °C overnight. The precipitated pellets were harvested by centrifugation at 12,000 g and 4 °C for 1.5 h. The pellets were resuspended using buffer B and passed through a 0.1-μm-pore filter twice to remove residual bacterial contaminations. The filtered resuspensions were then ultra-centrifuged at 150,000 g and 4 °C for 1 h. The pellets were resuspended in buffer B and filtered via 0.22 μm centrifugal filtering. The purified samples were analysed by negative staining EM, mass spectrometry and western blotting.

In the western blotting, the rabbit polyclonal antibodies against Alg1 (inner tube protein, generated from GenScript) or Alg2 (sheath protein, generated from GenScript) were respectively diluted to the final concentration of 1 μg ml⁻¹, while the rabbit polyclonal antibody against recA (abcam, ab63797) was regarded as the loading control and was diluted at the ratio 1:2,000.

**Mass spectrum analysis.** The purified AlgoCIS samples were sent in solutions to the Functional Genomics Center Zürich (FGCZ), which performed the mass spectrum and the subsequent data analysis. The samples were first digested by trypsin. These digested samples were dried and dissolved in 20 μl ddH₂O with 0.1% formic acid. The samples were transferred to autosampler vials for liquid chromatography–mass spectrometry analysis (LC–MS/MS). The samples were diluted at the ratio 1:40, with 1 μl of each sample being injected on a nanoAcquity UPLC coupled to a Q-Exactive mass spectrometer (ThermoFisher).

The acquired MS data were converted to a Mascot Generic File format and were processed for identification using the Mascot search engine (Matrixscience). In addition, the acquired MS data were imported into PEAKS Studio (Bioinformatic Solutions) and were searched against the *Algoriphagus machipongonensis* database. The results were visualized by Scaffold software.

**Vitrified sample preparations.** The purified AlgoCIS particles were vitrified on 200 mesh Quantifoil Gold grids (R 2/2) using a Vitrobot Mark IV (FEI company), whereas the contracted AlgoCIS particles were vitrified on Quantifoild Copper grids (R2/2) coated with 1 nm thickness of carbon layer. For cryoET, different samples (purified wild-type AlgoCIS and related mutants, wild-type bacteria at different OD₆₀₀s) were seeded with 10 nm BSA-coated colloidal gold particles at the ratio 1:5 before application to EM grids. The vitrified samples were first checked using a Tecnai F20 microscope (ThermoFisher) operating at 200 kV. The grids with appropriate ice thickness and good particle distribution were used for subsequent data collection.

Samples of MACs and *A. asiaticus* cells were prepared as previously described[16,36]. The samples were mixed with Protein A-conjugated 10 nm colloidal gold before plunge freezing using Quantifoil Copper grids (R2/1).

**CryoET data collection and tomogram reconstruction.** CryoET datasets of wild-type AlgoCIS and bacteria at different OD₆₀₀s were collected as movie stacks at a nominal magnification of 33,000 (an effective pixel size of 2.68 Å) with a defocus value of −8 μm. Tilt series collection was performed using a bidirectional scheme from −10° to +60° and then −12° to −60° in 2° incremental steps using SerialEM programme[51] on a Titan Krios EM (ThermoFisher) operating at 300 kV and equipped with an energy filter and a K3 Summit camera (Gatan). The tilt series of purified AlgoCIS mutants (AlgoCIS Cgo1/Cgo2⁻, Cgo1⁻, Cgo2⁻, and ΔAlg16B) were collected with the same bidirectional scheme at a nominal magnification of 53,000 (an effective pixel size of 2.75 Å) on a Titan Krios EM operating at 300 kV and equipped with an energy filter and a K2 Summit camera (Gatan). All datasets were collected at a defocus value of −8 μm. The dose rate of each tilt was ~2.1 e⁻/Å²

and the total dose was ~130 e⁻/Å². The tomograms were aligned, reconstructed and segmented using the IMOD programme suite[52]. The contrasts of some tomograms were further enhanced using the deconvolution filter 'tom_deconv'[53].

CryoET data of MACs were collected on a Titan Krios EM equipped with an energy filter and a K2 Summit direct electron detector. *A. asiaticus* data were collected using a K3 direct electron detector. Tilt series were acquired with the software SerialEM using a dose-symmetric tilt scheme. The angular range was −60° to +60° and the angular increment was 3°. The total electron dose varied between 120 and 150 e⁻/Å² and the pixel size at specimen level was 1.4 Å per pixel for MAC array data and 2.69 Å per pixel for *A. asiaticus* data. Images were recorded at a defocus value of −2 to −3 μm for MAC array data and −2 to −5 μm for *A. asiaticus* data. Frame alignment and dose weighting were performed using IMOD. Contrast transfer function (CTF) estimation was done using Gctf[54] and CTF correction was done by phase flipping in IMOD.

**Sub-tomogram averaging of wild-type AlgoCIS and mutants.** The sub-tomogram averaging of wild-type AlgoCIS, Cgo1/Cgo2⁻ and ΔAlg16B was performed with 'Dynamo'[55]. The individual AlgoCIS particles were manually picked using dipole set models from the reconstructed tomograms without CTF correction at a binning factor of 4. The sub-volumes of AlgoCIS particles (wild-type: 1,976 from 23 tomograms, Cgo1/Cgo2⁻: 404 from 46 tomograms, ΔAlg16B: 83 from 30 tomograms) were extracted and subsequently averaged. All sub-tomograms were split into half-datasets on the basis of the odd-and-even with 'dteo' package in 'Dynamo'. The half-datasets were individually aligned against the same reference for 2 iterations of coarse alignment and the subsequent 2 iterations of fine alignment, assuming 3-fold symmetry. The averaged sub-volumes from half-datasets were used to estimate the Fourier shell correlation using 'relion_postprocess'. The final resolutions of averaged sub-volumes of wild-type AlgoCIS, Cgo1/Cgo2⁻ and ΔAlg16B were estimated at 44 Å, 30 Å and 33 Å, respectively.

To generate the difference map between the averaged sub-volumes of wild-type and Cgo1/Cgo2⁻ mutant, the two volumes were aligned against each other and lowpass filtered to 45 Å. The difference map was further generated by volume subtraction of the wild-type against the Cgo1/Cgo2⁻ mutant using 'diffmap' (https://grigoriefflab.umassmed.edu/diffmap).

**Sub-tomogram averaging of the baseplate in MACs and *A. asiaticus*.** Dynamo was used for all sub-tomogram averaging steps. Particles were picked manually (including in plane orientation) from 110 and 68 tomograms of MACs and *A. asiaticus* cells, respectively. A total of 816 MACs and 366 *A. asiaticus* baseplate particles were extracted from tomograms binned by a factor of 4. An initial reference was obtained by averaging all particles using orientations from picking and applying 6-fold symmetry. After initial alignment limited to 40–50 Å, the dataset was split in two half-datasets using 'dteo' and processed independently. Particles were re-extracted from binned tomograms by a factor of 2 and alignment was further refined. The final resolution of the baseplate in MACs and *A. asiaticus* was estimated at 11 Å and 29 Å, respectively.

**CryoEM single particle data collection and image processing.** CryoEM datasets of purified AlgoCIS and contracted AlgoCIS were collected as movie stacks at a nominal magnification of 81,000 (an effective pixel size of 0.55 Å at super-resolution) using the SerialEM programme on a Titan Krios EM operating at 300 kV and equipped with an energy filter and a K3 Summit camera. The data collection was performed in super-resolution mode and the total exposure time was 1.5 s. Each stack contains 50 frames and the accumulated electron dose rate was ~60 e⁻/Å². The movie frames of each collected stack were aligned and summed up into one single micrograph with dose weighting at the binning factor of 2 using MotionCor2 (effective pixel size of 1.10 Å)[56]. The CTF parameter of the micrographs were estimated using Gctf. A total of 6,689 micrographs of purified AlgoCIS with good defocus value range (−1.0 to −3.0 μm) and low drift were selected for subsequent image processing, whereas a total of 7,984 micrographs of contracted AlgoCIS were used to determine the structure of the baseplate.

The image processing of purified AlgoCIS was performed as previously reported[24,25,30]. The AlgoCIS particles were picked manually using Relion3.0[57], with the start–end coordinate pairs in arbitrary direction (start point: cap module; end point: baseplate). The particle extraction was performed in 'Extract helical segments' mode to extract helical segments of AlgoCIS particles on the basis of the arbitrary start–end coordinate pairs. The first segments of AlgoCIS particles (128,473 particles), corresponding to the cap, were used for the structural determination of the cap module, whereas the last segments of AlgoCIS particles (128,473 particles) were applied to determine the baseplate complex. The middle segments (856,113 particles) were subjected to the structural determination of the extended sheath-tube module using helical reconstruction in Relion3.0[58] (Extended Data Fig. 2a).

For the cap module, the bad particles were first excluded through 2D classification. A total of 65,059 particles were then used for 3D refinement in 'helical reconstruction' mode without imposing helical parameters, where the prior tilt and psi information of each start–end coordinate pair were applied in the calculation. The initial reference was generated by scaling and lowpass filtering of the homologue PVC model (EMDB entry: 9763), and was used to generate the

initial AlgoCIS cap module reconstruction at the binning factor of 4, assuming 6-fold symmetry. The reasonable initial reconstruction was then subjected to the following 3D auto-refinement against the raw particle images. The CTF parameters of individual particles were optimized at the last stage of the refinement using 'relion_ctf_refine'. The structure of the AlgoCIS cap module was determined from 65,059 particles at the final resolution of 2.5 Å, assuming 6-fold symmetry (Extended Data Figs. 2a and 3a,b).

For the baseplate module, image processing was similar to that of the cap module (Extended Data Fig. 2a). The structure of the AlgoCIS baseplate was determined from 82,969 particles at the final resolution of 2.7 Å, assuming 6-fold symmetry. However, the map quality of the peripheral surface of the baseplate was poor for structural interpretation. Thus, an additional 3D classification focused on the baseplate was applied to improve the map quality. The particles in the best 3D class (Class VIII) were used for the 3D focused refinement, resulting in a 2.9 Å resolution structure of the baseplate complex (Extended Data Figs. 2a and 3a,b). Since there is a symmetry mismatch in baseplate (C1 + C3 + C6), a symmetry relaxation was performed from 6-fold to 3-fold[59] (Extended Data Fig. 2a). Briefly, the symmetry equivalent orientations were first generated by adding an additional 60° to '_rlnAngleRot' of individual particles, and combined with the corresponding original orientations. Then, the 3D classification without angular sampling was performed among these combined orientations with a mask around the central spike. The particles with the correct 3-fold orientations were clearly separated in this process and subjected to a local 3D refinement. The final structure of 2.8 Å resolution of the baseplate was reconstructed from 82,696 particles with applied 3-fold symmetry (Extended Data Fig. 3a,b).

For the overall AlgoCIS structure, the centre of the individual baseplate was moved along the z axis (or 3-fold symmetry axis) to the centre of the AlgoCIS particle on the basis of the refined orientation. The AlgoCIS particles were then re-extracted from raw micrographs and the relative orientation of individual baseplates was applied to the corresponding CIS particles. The bad particles were removed through 2D classification and the good particles were applied to a local 3D refinement at the binning factor of 2, assuming 3-fold symmetry. A total of 52,837 particles were used to determine the 4.4 Å structure of the overall AlgoCIS particle (Extended Data Fig. 2a). Although the resolution of the AlgoCIS particle reached the Nyquist frequency at the binning factor of 2, the structure of AlgoCIS was sufficient for the structural assignment of the entire atomic model. In addition, the big box size of particles (1,600 pixels) would occupy a lot of computational resources. Thus, we did not continue the structural refinement against the raw images at the binning factor of 1.

For the extended sheath-tube module, the middle segments were subjected to two rounds of 2D classification to discard the bad particles at the binning factor of 4. The particles in good classes were first subjected to 3D auto-refinement using helical reconstruction without applying helical parameters at the binning factor of 4. The initial helical parameters were deduced from the initial reconstruction in real space using 'relion_helix_toolbox' and then optimized in the subsequent 3D helical refinement. The final 2.4 Å resolution structure of the extended sheath-tube module was determined from 225,305 particles calculated with 6-fold symmetry and helical parameters (rise = 40.80 Å, twist = 20.54°) (Extended Data Figs. 2a and 3a,b).

For the contracted sheath, the helical particles were first manually picked using start–end pairs in Relion3.0 and we then performed particle extraction in 'Extract helical segments' mode. The extracted helical segments were subjected to 2D classification and the particles in good classes were subjected to the same helical processing as that of the extended sheath-tube module (Extended Data Fig. 2b). The final 2.5 Å resolution structure of contracted sheath was obtained from 92,922 particles applied with 6-fold symmetry and helical parameters (rise = 18.04 Å, twist = 32.20°) (Extended Data Fig. 3c,d).

To determine the baseplate of contracted AlgoCIS, the baseplate particles were manually picked in Relion3.0 (Supplementary Fig. 2a). A total of 21,933 particles were extracted and directly subjected to 3D auto-refinement, assuming 6-fold symmetry with the 60 Å lowpass-filtered baseplate model in contracted AFP (EMDB entry: 4876) at the binning factor of 4. Local 3D classification was applied to the baseplate part. The particles from bad 3D classes (class I–III) were subjected to one additional 2D classification without angular sampling and the good particles were combined with the particles in 3D classes IV and V. Then the combined particles were subjected to 3D auto-refinement at the binning factor of 2. The density around the baseplate was further refined by three rounds of focused 3D classification. The particles in three rounds of good classes were combined and the duplicates were removed. A total of 3,793 particles were used to determine the 4.8 Å resolution structure of the overall contracted AlgoCIS, which was mainly attributed to the map quality of the sheath-tube module. The resolution of the baseplate was estimated via a local mask, which was resolved to 8.9 Å resolution (Supplementary Fig. 2b).

The resolutions of relative reconstruction maps were estimated on the basis of the gold-standard Fourier Shell Correlation (FSC) = 0.143 criteria[60]. The local resolution estimations for individual maps were performed using the local resolution module in Relion3.0 and examined using UCSF Chimera[61] (Extended Data Fig. 3 and Supplementary Fig. 2c).

**Structural modelling of AlgoCIS.** The map qualities of different modules of AlgoCIS (cap, extended sheath tube and baseplate) and the contracted sheath allowed for de novo structural modelling (Extended Data Fig. 4). The atomic models of Alg16A, Alg16B, Alg2, Alg1, Alg5, Alg6, Alg7, Alg8, Alg9, Alg11 and Alg12 were manually built in COOT[62]. The models were then subjected to iterative real-space refinements against related density maps using RosettaCM[63] and 'phenix.real_space_refine'[64]. The final models were evaluated using 'phenix.molprobity'[64] (Supplementary Table 3) and the correlations between models and the corresponding maps were estimated using 'phenix.mtriage'[64] (Extended Data Fig. 3a,c).

Out of 18 predicted open reading frames in the AlgoCIS gene cluster, 7 proteins (Alg16A, Alg16B, Alg1, Alg5, Alg7, Alg8 and Alg9) were built with almost full-length atomic models in the final structure. For the sheath protein Alg2, the density on the distal sheath layer (Alg2-L$_{23}$) was the best one to build the atomic model. However, the densities of some loops in domain VI of Alg2 (residues 288–320) were still poor and did not allow for structural modelling, and the residues 430–446 of domain VI were only assigned with the poly-alanine chain. The disordered loop of Alg6 (residues 19–28) was missing in the final structure. Due to the poor density on the peripheral part of the baseplate, the final model of Alg12 includes gp6-like part and part of gp7-like part (residues 3–551). The density of the domain VI of Alg11 (residues 916–1,012) allowed for the tracing of the main chain, but was not good enough for the assignment of the side chains. Thus, only a poly-alanine chain was built in this part.

As the resolution of the baseplate in contracted AlgoCIS was not good enough for structural modelling (Supplementary Fig. 2c), the final model was generated by rigid-body fittings of the baseplate wedge components (Alg11–12) in the extended AlgoCIS. Molecular graphs were made using Chimera and ChimeraX[65].

**Generation of AlgoCIS mutants.** The A. machipongonensis AlgoCIS deficient mutants generated in this paper are all site-directed insertion mutants with pYT313[66], a bacteroidetes suicide plasmid containing an erythromycin-resistant cassette, inserted into the genes of interest. A 2,300 bp long sequence containing the sequence of the gene of interest was cloned into pYT313. The constructed plasmid was then transformed into A. machipongonensis with a modified protocol[67]. Briefly, a 5 ml overnight bacterial culture was spun down and washed with sucrose buffer (272 mM sucrose, 1 mM MgCl$_2$, 7 mM K$_2$HPO$_4$, pH 7.5) three times and then sucrose buffer containing 15% (v/v) glycerol three times at 4 °C. The cells were then mixed with the extracted plasmids and loaded into a 1-mm-wide Gene pulser electroporation cuvette (Biorad). The electroporation was performed using Gene Pulser Xcell Electroporation System (Biorad) with 1.5 kV, 200 Ω resistance and 25 μF capacitance. Fresh MB medium was added and the cells were recovered at 30 °C for 4 h. The recovered cells were plated on marine (MB) agar plates (Condalab) containing 50 μg ml$^{-1}$ erythromycin to select mutants with pYT313 inserted. All bacterial mutants were further confirmed by colony PCR.

For AlgoCIS$^-$ mutant, 2,300 bp of the nucleotide sequence containing the N-terminal coding region of Alg8 was cloned and inserted into pYT313. For the null mutant, 2,300 bp of a pseudogene (ALPR1_RS19250) was cloned into pYT313 and the insertion of the constructed plasmid into the pseudogene did not disrupt any functional gene in the bacterial genome. For AlgoCIS Cgo1/Cgo2$^-$ and individual mutants (Cgo1$^-$ and Cgo2$^-$), the 2,300 bp upstream of the related gene, all the downstream genes in the AlgoCIS gene cluster and the erythromycin-resistant cassette, including the corresponding promoter, were synthesized and further constructed into pUC57 vector (produced from Genscript) (Extended Data Fig. 9b). The primers used for AlgoCIS$^-$ and null mutants are listed in Supplementary Table 4.

The in-frame deletions of the AlgoCIS genes were generated utilizing our pCHIP3 suicide plasmid (Extended Data Fig. 4e). Approximately 1,000 bp long DNA fragments flanking the target gene of interest were amplified via PCR and ligated with Gibson Assembly (New England Biolabs) into linearized plasmid. The assembled plasmid was then electroporated into SM10 E. coli, with the sequence of the assembled product being confirmed via DNA sequencing before being conjugated with A. machipongonensis on MB agar plates and incubated at 30 °C overnight. The following day, the mating spots were resuspended in MB and re-plated onto MB agar plates containing 100 μg ml$^{-1}$ erythromycin and grown for at least 48 h at 30 °C. Colonies that formed were confirmed to have the pCHIP3 plasmid inserted via PCR, with good clones being grown in MB for 48 h at 30 °C, shaking at 200 r.p.m. After incubation, 1 ml of the culture was spun down and washed twice with filtered artificial seawater before being plated on MB agar plates containing 10 mM 4-chloro-DL-phenylalanie and incubated at 30 °C for at least 72 h. Individual colonies that formed were then screened for loss of plasmid and confirmed for the gene deletion via PCR, as well as being sequenced to confirm the clean knockout. The related primers used for in-frame deletions are listed in Supplementary Table 4.

**Generation and purification of MAC mutants.** Mutants in P. luteoviolacea were created with pCVD443 suicide vectors following previously reported protocols[36]. MACs were purified following the established protocol[22,36]. Briefly, the P. luteoviolacea strains were grown in 50 ml artificial seawater with tryptone medium (SWT) at 30 °C, 200 r.p.m. overnight in 250 ml flasks. The cells were centrifuged for

20 min at 4,000 g and 4 °C, with the resulting supernatant being removed. The pellet was gently resuspended in 5 ml cold extraction buffer (20 mM Tris, 1 M NaCl, pH 7.5). The resuspension was then spun down again for 20 min at 4,000 g and 4 °C. The supernatant was carefully pipetted into a new 15 ml conical tube and spun down for 30 min at 7,000 g and 4 °C. After centrifugation, the supernatant was removed and the small pellet was resuspended in residual buffer and stored at 4 °C for use. The resuspended samples were subjected to killing assays and negative-stain EM imaging.

**Phylogenetic tree analysis.** The phylogenetic trees of different contractile injection systems were examined using the putative sheath proteins on the basis of previous reports[16]. The amino acid sequences were first aligned by the MUSCLE online tool[68,69] and further subjected to tree reconstruction in the MEGAX programme[70]. The maximum likelihood method and bootstrap values (1,000 resamples) were applied to assess the robustness of the tree.

**Genetic cloning and bacterial intoxication assays.** The full-length of the Cgo1 (ALPR1_12695) gene was PCR amplified from the bacterial genome. A His$_6$ tag was introduced to the C terminus of the recombinant Cgo1 through the reverse PCR primer. The PCR product was cloned into the pET-DUET (Novagen) vector by the Gibson assembly method or T4 ligation (New England Biolabs). The full-length protein fused with N-terminal periplasmic tag, different point mutants, and truncations of Cgo1 were subsequently generated on the basis of the recombinant Cgo1 construct. The full-length Cgo1 and related mutants were expressed in *E. coli* BL21 (DE3) cells. In addition, the recombinant Cgo2 (ALPR1_12690) gene, Cgo1/Cgo2 genes, and sfGFP gene were also expressed in *E. coli* BL21 (DE3) cells following the same procedures as that of Cgo1.

To evaluate the intoxication effects of recombinant proteins, the bacteria with the different constructed vectors were grown in LB medium overnight at 37 °C. The overnight bacterial cultures were adjusted to $OD_{600} = 3.0$ and serially diluted using LB medium. The 5 μl volumes of serial dilutions were spotted on LB agar containing related antibiotic (100 μg ml⁻¹ ampicillin) and without/with inducer (0.1 mM IPTG). The bacteria carrying the empty vector (pET-DUET) and sfGFP protein were regarded as negative controls.

**Bacterial two-hybrid assays.** A bacterial two-hybrid analysis was used to investigate protein–protein interactions following protocols detailed previously[71]. Briefly, proteins of interest were cloned into one of four bacterial two-hybrid (BACTH) plasmids containing the T18 or T25 subunits of the adenylate cyclase protein. Plasmid fusion combinations were transformed into BTH101 *E. coli* electrocompetent cells. The BTH101 cells were grown on LB agar containing ampicillin (100 μg ml⁻¹), kanamycin (100 μg ml⁻¹) and 1% glucose to suppress expression. Protein–protein interactions were quantified by performing a β-galactosidase assay, with cells being grown overnight at 37 °C and shaking at 200 r.p.m. Protein expression was induced with 1.0 mM IPTG for 8 h at 30 °C. The induced cells were then mixed with a one-step 'β-gal' mix[72]. A plate reader was then used to measure the absorbance at 420 nm and 600 nm. The values were then used to calculate Miller Units as previously described[73].

**Killing assays of AlgoCIS against bacteria.** To estimate whether AlgoCIS particles could target bacteria, different bacterial strains (*E. coli* BL21 (DE3) with pET-Duet vector, *Vibrio cholerae* and *Echinicola pacifica*) were used in bacterial killing assays with purified wild-type AlgoCIS and Cgo1/Cgo2⁻ samples. Briefly, the different bacterial strains were inoculated into the corresponding medium and grown overnight. The next day, the cultures were adjusted to $OD_{600} = 0.2$ by the corresponding medium and then mixed with the equal volume of 0.25 mg ml⁻¹ purified wild-type AlgoCIS or Cgo1/Cgo2⁻, whereas the buffer (20 mM Tris pH 7.5, 150 mM NaCl) was regarded as the negative control. The mixtures were further incubated for one additional day on shaker and serial dilutions on agar plates were then performed.

**Killing assays against different types of eukaryotic cells.** Before treatments, eukaryotic cell strains were grown in multi-well dishes to a confluency of around 70–80% at their respective temperatures and growth conditions (Sf9 at 28 °C, J774A.1 at 37 °C and 5% CO₂, *A. castellanii* at 37 °C, and *D. discoideum* at 22 °C), with n = 4 for each assay. Purified protein samples (AlgoCIS or MACs) or a bacterial culture ($OD_{600} = 3.5$) were diluted at 1:10/1:25/1:50, respectively, with the eukaryotic cell growth media. The medium in the multi-well dish was removed and replaced with the AlgoCIS/MAC (or bacterial dilution)-containing media and the plate was incubated back at growth temperatures. The plates were observed for change at 24 h, 48 h and 72 h post treatments. Cells were live/dead stained with FDA/PI and images were taken of each well via fluorescent light microscopy on an inverted Zeiss 200M fluorescent microscope with a 20×0.4 NA Korr Ph2 LD Plan Neofluar objective using Metamorph (version 7.7.11.0).

To screen whether choanoflagellates were sensitive to AlgoCIS, *S. rosetta* cell proliferation was determined by diluting cultures to a concentration of 10⁴ cells per ml in high-nutrient media and distributing 1 ml of culture into each well of a 24-well plate. Purified wild-type AlgoCIS and Cgo1/Cgo2⁻ (5 μg ml⁻¹, 1 μg ml⁻¹, 500 ng ml⁻¹, 100 ng ml⁻¹) were added, whereas heat denaturation (95 °C for 30 min) and freeze–thawing were regarded as the negative controls. At different time points

(21 h, 28 h, 45 h, 52 h and 69 h), 100 μl from each well were transferred and fixed with 5 μl 37.5% formaldehyde by vortexing. The fixed cells were stored at 4 °C until the sample was used for determining cell concentration, which was done by counting the number of cells in a fixed-volume imaging chamber from three distinct regions of the chamber[74].

The *S. rosetta* cells were isolated from 30 ml *S. rosetta* culture with prey bacteria *E. pacifica* (SrEpac) via Percoll density gradient centrifugation. The *S. rosetta* cell pellet was resuspended in artificial (AK) seawater[75] and the residual *E. pacifica* bacteria were killed by 2 d treatment with an antibiotic cocktail (200 μg ml⁻¹ lincomycin, 160 μg ml⁻¹ erythromycin and 20 μg ml⁻¹ rifampicin). The antibiotic-treated *S. rosetta* culture was centrifuged. The cell pellets were resuspended in 10 ml high-nutrient media, ready for bacterial treatments[75].

During the antibiotic treatment of the choanoflagellate culture, the colony of wild-type *A. machipongonensis*, Cgo1/Cgo2⁻ and AlgoCIS⁻ mutants were grown in seawater (24 g l⁻¹ Tropic Marine Salts, 5 g l⁻¹ bacto peptone, 3 g l⁻¹ yeast extract, 0.3% v/v glycerol) with and without 50 μg ml⁻¹ erythromycin at 22 °C for 2 d. The bacterial cultures were pelleted and resuspended in high-nutrient media[75] to a final concentration of 20 mg ml⁻¹. The bacterial resuspensions were added into the *S. rosetta* culture at the ratio 1:1,000, while an additional 50 μg ml⁻¹ erythromycin was added for Cgo1/Cgo2⁻ and AlgoCIS⁻ mutants. After 24 h, the *S. rosetta* culture was taken and rosette integrity was assessed after vortexing. The *S. rosetta* cell density and rosette formation were further analysed by wheat germ agglutinin (WGA), staining the localization of rosetteless protein[76].

All images were further analysed and processed with Fiji[77] and all plots were processed with Prism 8.0.

**Reporting Summary.** Further information on research design is available in the Nature Research Reporting Summary linked to this article.

## Data availability

The cryoEM density maps and corresponding atomic models have been deposited in the EMDB and PDB, respectively. The accession numbers are listed as follows: cap module of AlgoCIS (PDB: 7ADZ and EMD-11734); sheath-tube module in the extended state (PDB: 7AE0 and EMD-11735); baseplate reconstructed in C6 symmetry (EMD-11743); baseplate focused refinement in C6 symmetry (PDB: 7AEB and EMD-11744); baseplate reconstructed in C3 symmetry (PDB: 7AEF and EMD-11745); overall AlgoCIS reconstruction (EMD-11746); sheath-tube module in the contracted state (PDB: 7AEK and EMD-11747); baseplate in the contracted state (EMD-11748). The representative reconstructed tomograms and the sub-tomogram averages have been deposited in the EMDB. The accession numbers are listed as follows: sub-tomogram average of purified AlgoCIS (EMD-11749); sub-tomogram average of purified AlgoCIS Cgo1/Cgo2⁻ mutant (EMD-11750); sub-tomogram average of purified AlgoCIS ΔAlg16B (EMD-13723); sub-tomogram average of baseplate in MAC arrays (EMD-13724); sub-tomogram average of baseplate in *A. asiaticus* (EMD-13725); in situ reconstructed tomogram of *A. machipongonensis* in bacterial late stage (EMD-13705); reconstructed tomogram of purified AlgoCIS (EMD-13722); reconstructed tomogram of purified AlgoCIS Cgo1/Cgo2⁻ mutant (EMD-13719); reconstructed tomogram of purified AlgoCIS Cgo1⁻ mutant (EMD-13720); reconstructed tomogram of purified AlgoCIS Cgo2⁻ mutant (EMD-13721); reconstructed tomogram of purified AlgoCIS Δalg16B (EMD-13704).

The corresponding PDB entries (1OF4, 4JX0, 4ZXE, 5MWN, 6J0M, 6J0F, 6J0B, 6J0N, 6RBN, 6RBK, 6RAP, 6RAO) were used for the structural superpositions in the manuscript. The EMDB entries (EMD-9763, EMD-4876) were lowpass filtered and were used as initial reference in the structural determination procedure.

The raw numerical data for Figs. 2d and 5a, and Extended Data Figs. 9c and 10d are included in the source data. The Newick files for the phylogenetic trees in Figs. 1b and 6a are provided in the source data.

Due to the size of the reconstructed tomograms of different AlgoCIS mutants, all tomograms are available from the authors on reasonable request. Source data are provided with this paper.

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

## Acknowledgements

We thank ScopeM for instrument access at ETH Zürich; the Functional and Genomic Center Zürich (FGCZ) for mass spectrometry support; Pilhofer Lab members for discussions; M. J. McBride (University of Wisconsin-Milwaukee) and H. Hilbi (University Zürich) for providing plasmids and cells. M.P. was supported by the Swiss National Science Foundation (31003A_179255), the European Research Council (679209) and the NOMIS foundation.

## Author contributions

J.X. and M.P. conceived the project; J.X. performed the sample preparation, processed the cryoEM data, reconstructed the cryoEM map, built and refined the structural model; C.F.E. developed the bacterial knockout system and performed genetic analysis; C.F.E. performed bacterial two-hybrid assays; Y.-W.L. and J.X. generated bacterial deficient mutants; J.X. and M.F. collected cryoET data and analysed the reconstructed tomograms; J.X. and F.E. performed the sub-tomogram averaging; C.F.E. and J.X. performed killing assay screenings; F.U.N.R. performed assays on choanoflagellates; J.X., C.F.E. and M.P. wrote the manuscript; all authors commented on the manuscript.

## Competing interests

C.F.E and M.P. have two provisional patents related to CIS pending in the United States (application no. 62/768,240 and 62/844,988). All other authors declare no competing interests.

## Additional information

**Extended data** is available for this paper at https://doi.org/10.1038/s41564-022-01059-2.

**Correspondence and requests for materials** should be addressed to Martin Pilhofer.

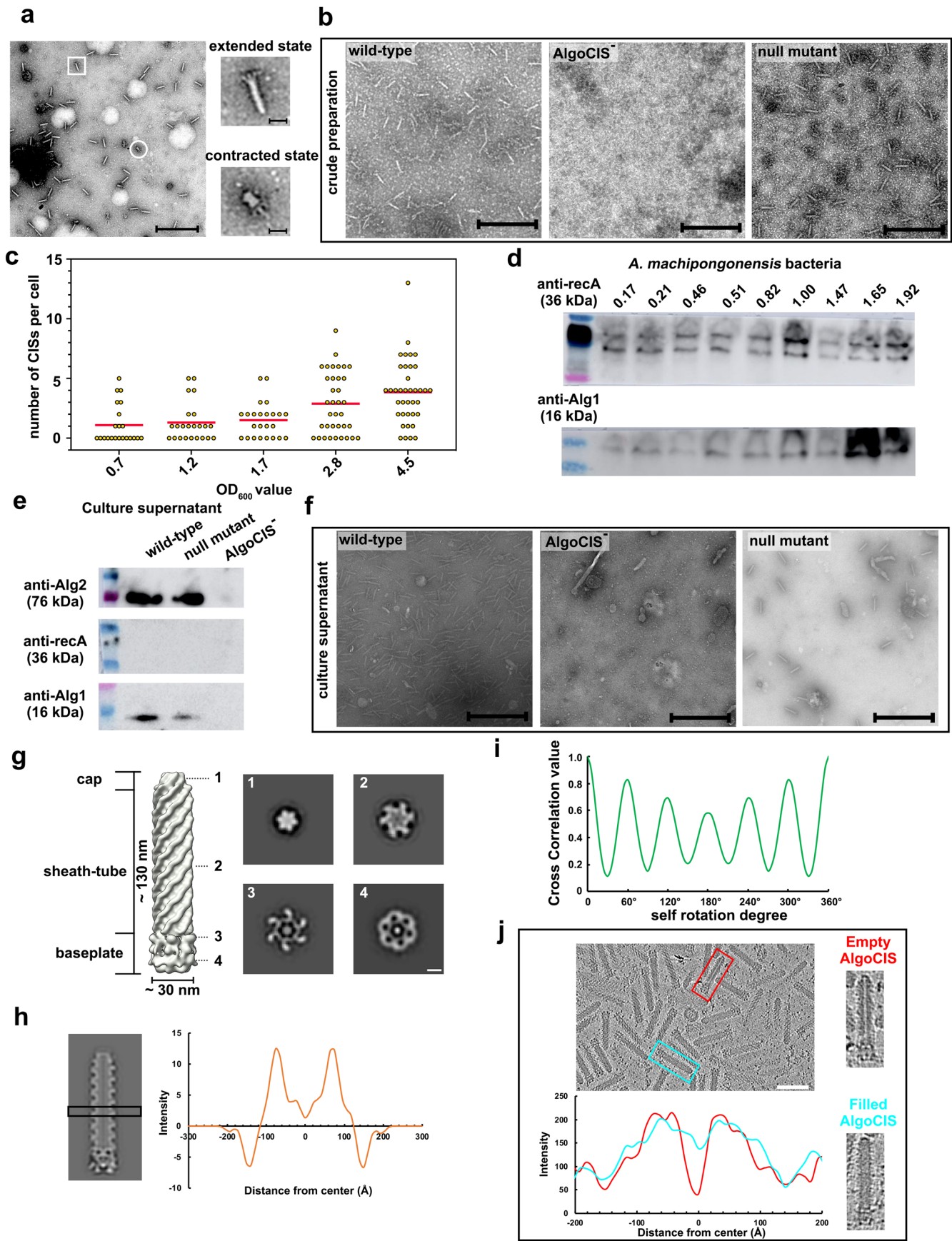

**Extended Data Fig. 1 | See next page for caption.**

**Extended Data Fig. 1 | Biochemical and EM characterization of AlgoCIS. a:** Negative staining EM image of crude sheath preparation from *A. machipongonensis* showing the typical CIS-like particles in both extended (box) and contracted (circle) states. Bar: 500 nm. The magnified views of the representative particles are shown in the right panels (bars: 50 nm). This experiment was performed ten independent times. **b:** Negative staining EM images of crude sheath preparations from different *A. machipongonensis* bacteria (wild-type, AlgoCIS⁻, and null mutant). Bars: 500 nm. This experiment was performed three independent times. **c:** Plot showing the number of AlgoCISs per cell as observed by cryoET (averages shown as red lines). AlgoCISs were most abundant at high optical densities ($OD_{600}$). **d:** Western blotting analyses of different bacterial samples against the putative inner tube protein (Alg1), showing that the expression level of AlgoCIS is high at high bacterial $OD_{600}$. The cytosolic protein recA is used as loading control. This experiment was performed three independent times. **e:** Western blotting analyses of the purified samples from different bacterial culture supernatants. Both putative sheath (Alg2) and inner tube (Alg1) proteins could be detected in the bacterial cultures from wild-type and null mutant, while neither protein could be detected in AlgoCIS⁻ mutant. This experiment was performed three independent times. **f:** Negative staining EM images of the AlgoCIS purifications from bacterial culture supernatant in different *A. machipongonensis* bacteria (wild-type, AlgoCIS⁻, and null mutant). Bars: 500 nm. This experiment was performed three independent times. **g:** Sub-tomogram average revealing three AlgoCIS structural modules: cap, sheath-tube, and baseplate complex. Corresponding volume slices (perpendicular views) are shown on the right. Bar: 10 nm. **h:** Central volume slice (left) of sub-tomogram average of AlgoCIS and the corresponding density plot (right) showing that the inner tube lumen has density. **i:** Self-rotation plot showing that the sub-tomogram averaging of AlgoCIS has 6-fold symmetry features even though 3-fold symmetry was applied in data processing. **j:** Representative cryoET slice of the purified wild-type AlgoCIS showing the inner tube lumen is filled with density in most particles. The representative filled and empty AlgoCIS particles are highlighted by boxes (filled: cyan; empty: red) and are zoomed in in the right panels. Shown is a 10.8 nm thick slice. Bar: 50 nm. The corresponding density plots of the filled (cyan) and empty (red) AlgoCIS particles are shown in the bottom panel, where the symmetry axis is set as the center. The density measurements were performed independently in three different reconstructed tomograms.

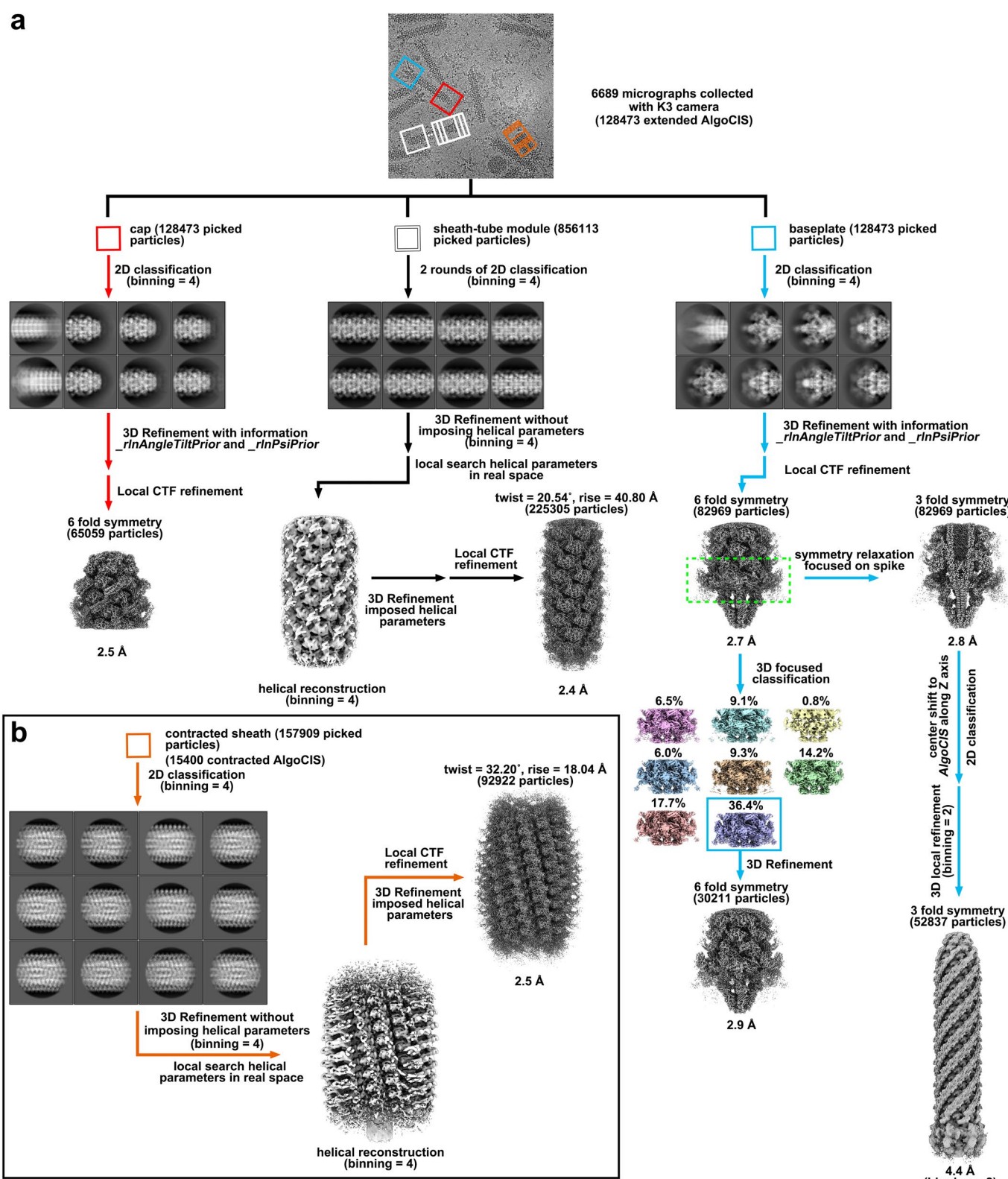

**Extended Data Fig. 2 | Workflows for the cryoEM structural determinations of extended AlgoCIS particle and contracted sheath. a:** Flowcharts for the cryoEM reconstructions of the different modules of AlgoCIS particle in the extended state, including cap (red), sheath-tube module (white/black), baseplate complex (blue), and the overall AlgoCIS particle. See METHODS and Supplementary Table 3 for details. **b:** Flowcharts for the cryoEM reconstruction of the contracted AlgoCIS sheath (orange). See METHODS and Supplementary Table 3 for details.

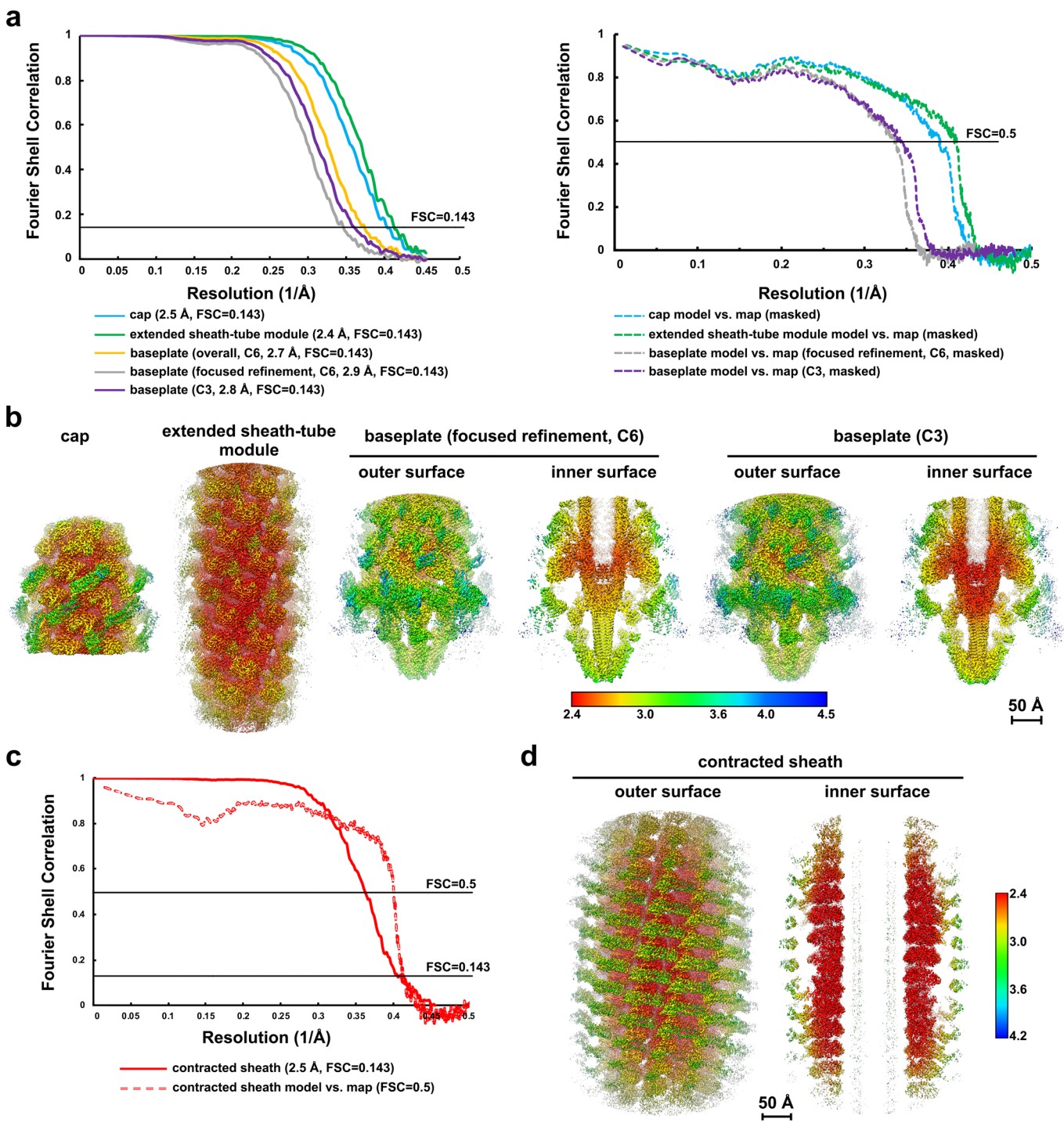

**Extended Data Fig. 3 | CryoEM analyses of reconstruction maps and models. a:** Gold standard Fourier shell correlation (FSC) curves (left) of the reconstructions of different modules in the extended AlgoCIS particle. The FSC of model *vs.* map (right) showing the structural qualities of different AlgoCIS modules. **b:** Local resolution maps of the reconstructions of different modules in the extended AlgoCIS. Bar: 50 Å. **c:** Gold standard FSC and model *vs.* map FSC curves of the contracted AlgoCIS sheath. **d:** Local resolution maps of the contracted AlgoCIS sheath. Bar: 50 Å.

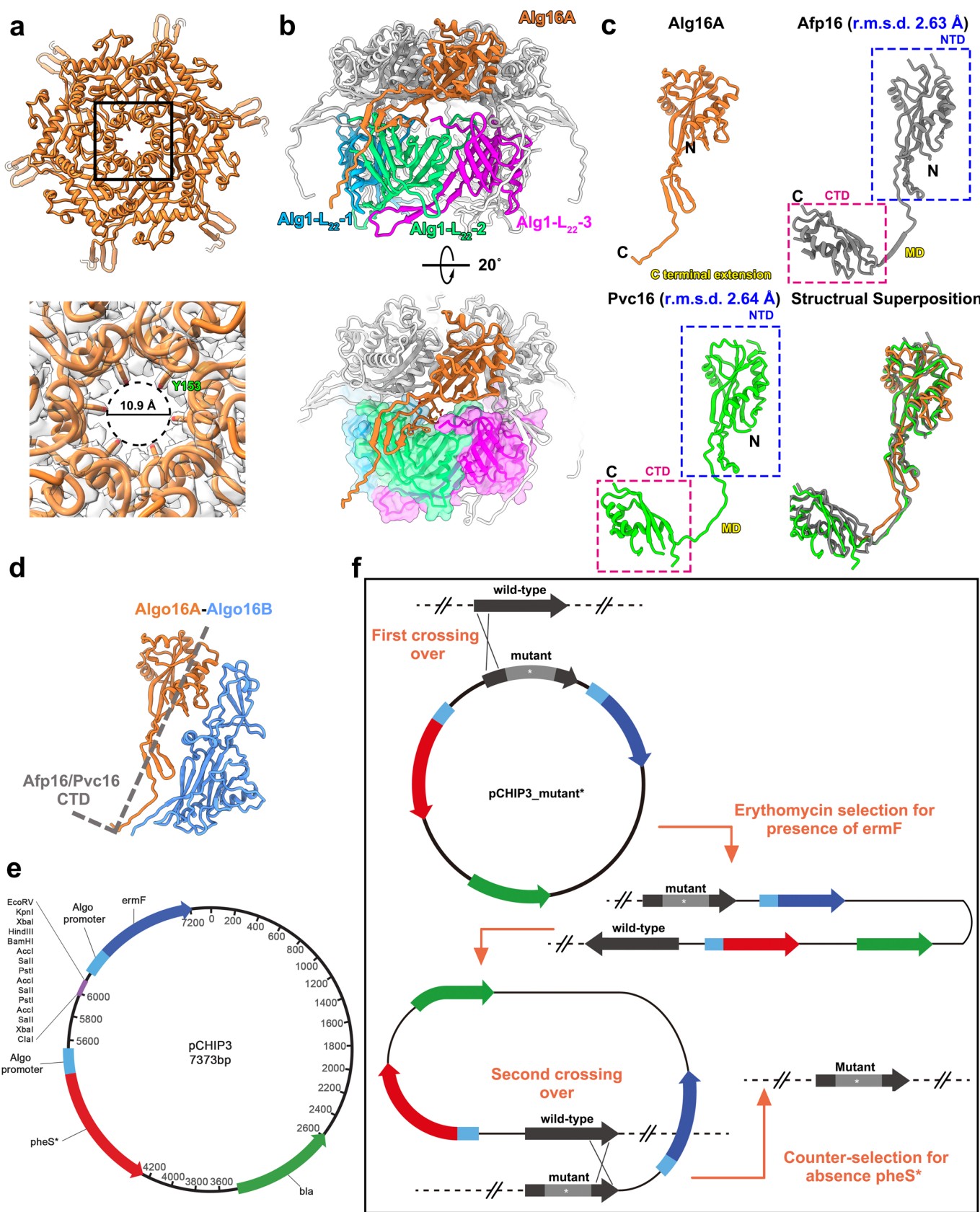

**Extended Data Fig. 4 | See next page for caption.**

**Extended Data Fig. 4 | Structural analyses of the cap module and clean in-frame bacterial genetic manipulation in AlgoCIS. a:** Shadowed surface and ribbon diagrams of the top view of the hexameric cap protein (orange). Box indicates the position of the central channel of the hexameric cap protein, which is shown in the bottom panel. The side chain of Tyr153 residues (Y153) of the central helix from each subunit points inwards, forming the central channel with a diameter of 10.9 nm. **b:** Shadowed surface and ribbon diagrams showing the interactions between cap protein and the distal layer of the inner tube. One cap protein (orange) has interactions with three inner tube proteins (Alg1-L$_{22}$-1: blue; Alg1-L$_{22}$-2: green; Alg1-L$_{22}$-3: magenta). The residues of Alg16A participating in contacts are shown with side chains in the bottom panel. **c:** Ribbon diagrams showing the structures of the cap protein (Alg16A: orange) and the homologous proteins (Afp16: grey; Pvc16: green). The individual domains in homologs are marked with dashed boxes in different colors (NTD: blue; CTD: magenta). The calculated structural R.M.S.D values of Alg1 against NTD domains in the homologs are indicated. The structural superimpositions of Alg16A and homologs are shown on the bottom right. **d:** Ribbon diagram showing that Alg16B (blue) stands parallel against Alg16A (orange), while the CTD domains of Afp16/Pvc16 (grey dashed line) adopt vertical arrangement against NTD domains. **e:** Schematic showing detailed information of the vector pCHIP3, which was used for bacterial clean in-frame deletion. **f:** Schematic showing the workflow of bacterial clean in-frame genetic manipulation in *A. machipongonensis* using pCHIP3. Please refer to METHODS for details.

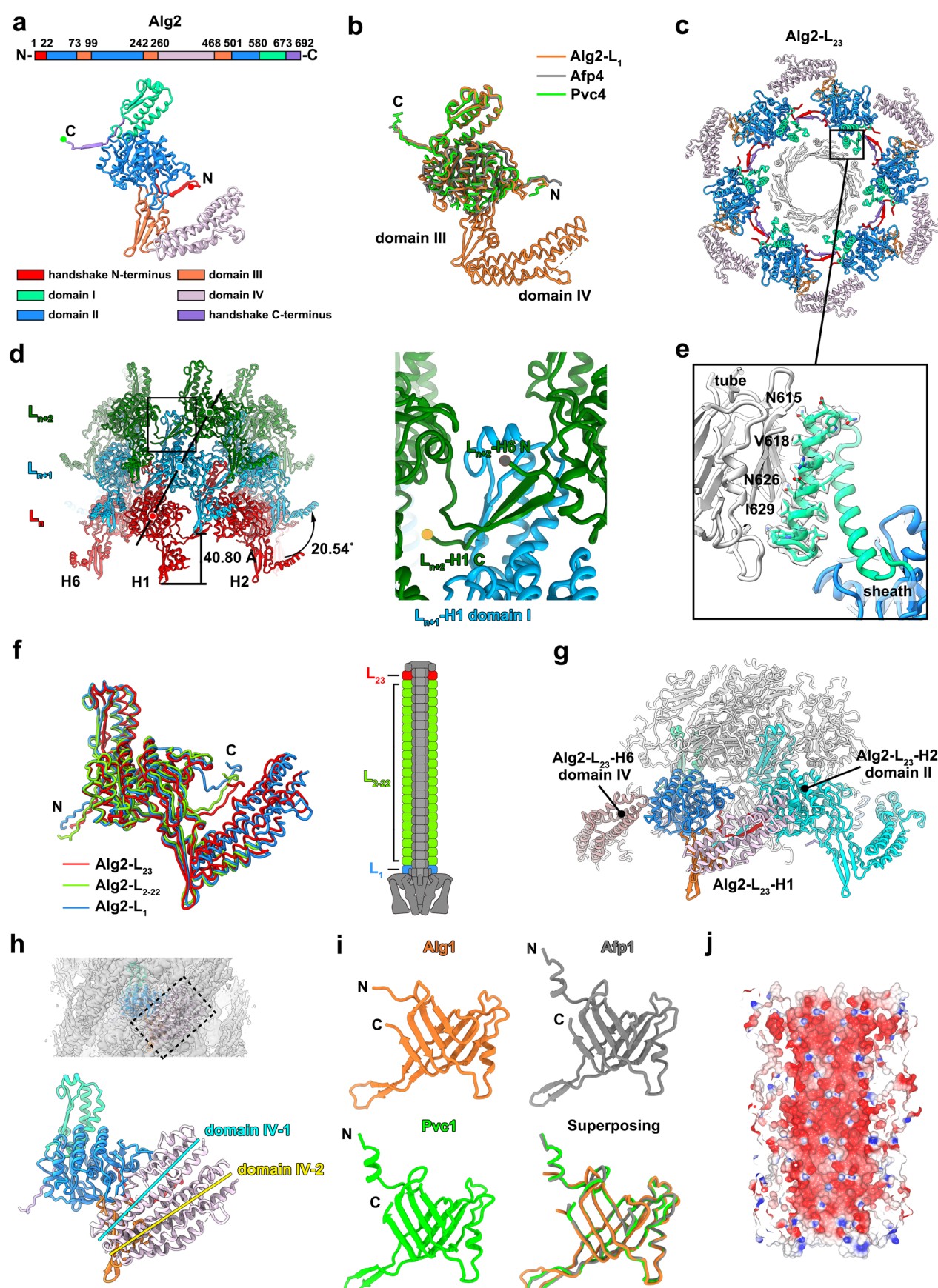

**Extended Data Fig. 5 | See next page for caption.**

**Extended Data Fig. 5 | Structural analyses of sheath-tube module in extended state. a:** Schematic and ribbon diagrams showing that the sheath protein (Alg2) has four domains. The N- and C-termini are marked with red and green circles. **b:** Structural superpositions of sheath protein (Alg2: orange) with homologous proteins (Pvc4: green; Afp4: grey). The additional domains (domain III and IV), N-, and C-termini of Alg2 are labeled. **c:** Ribbon diagram of a perpendicular slice of the distal sheath layer (Alg2-$L_{23}$), showing the domain organization of Alg2 in one sheath layer and the interactions between the sheath and inner tube (white). The color code matches panel **a**. Box indicates the interface between sheath and inner tube, which is shown in panel **e**. **d:** Ribbon diagrams showing an extended sheath fragment containing three sheath layers ($L_n$, $L_{n+1}$ and $L_{n+2}$) in different colors. The direction of one helical strand (H1) is highlighted by a line. Box indicates the conserved handshake interaction between sheath subunits, which is shown in the right panel. **e:** Shadowed surface and ribbon diagram showing the interactions between the attachment helix of sheath and inner tube. The residues mediating contacts are labeled and shown with side chains. **f:** Structural superpositions of sheath subunits from different layers (distal layer: red; central layers: green; proximal layer: blue) showing the structural diversity in domain IV, N-, and C-termini of Alg2 across different sheath layers. The schematic (right) indicates the position of the different sheath layers on the AlgoCIS particle. **g:** Side view of ribbon diagram showing the overall structure of the cap module and the distal sheath layer. The inner tube proteins, cap, and cap adaptor are colored white. The color code for different domains of one Alg2 subunit (Alg2-$L_{23}$-H1) matches panel **a**, whereas the neighboring two subunits are shown in different colors (Alg2-$L_{23}$-H2: cyan, Alg2-$L_{23}$-H6: brown). **h:** Shadowed surface and ribbon diagrams showing that there are two conformations of domain IV (domain IV-1, 2) in the central sheath layers. The color code for different domains matches panel **a**, while two conformations of domain IV are represented by lines with different colors (domain IV-1: blue; domain IV-2: yellow). **i:** Ribbon diagrams showing the structures of inner tube protein (Alg1: orange) and the homologs (Afp1: grey; Pvc1: green). The structural superpositions of Alg1 with homologs are shown in the bottom right panel. **j:** Surface electrostatic potential of the inner tube showing that the inner surface of tube is dominated by negative charges. Negative and positive electrostatic potentials are colored red and blue, respectively.

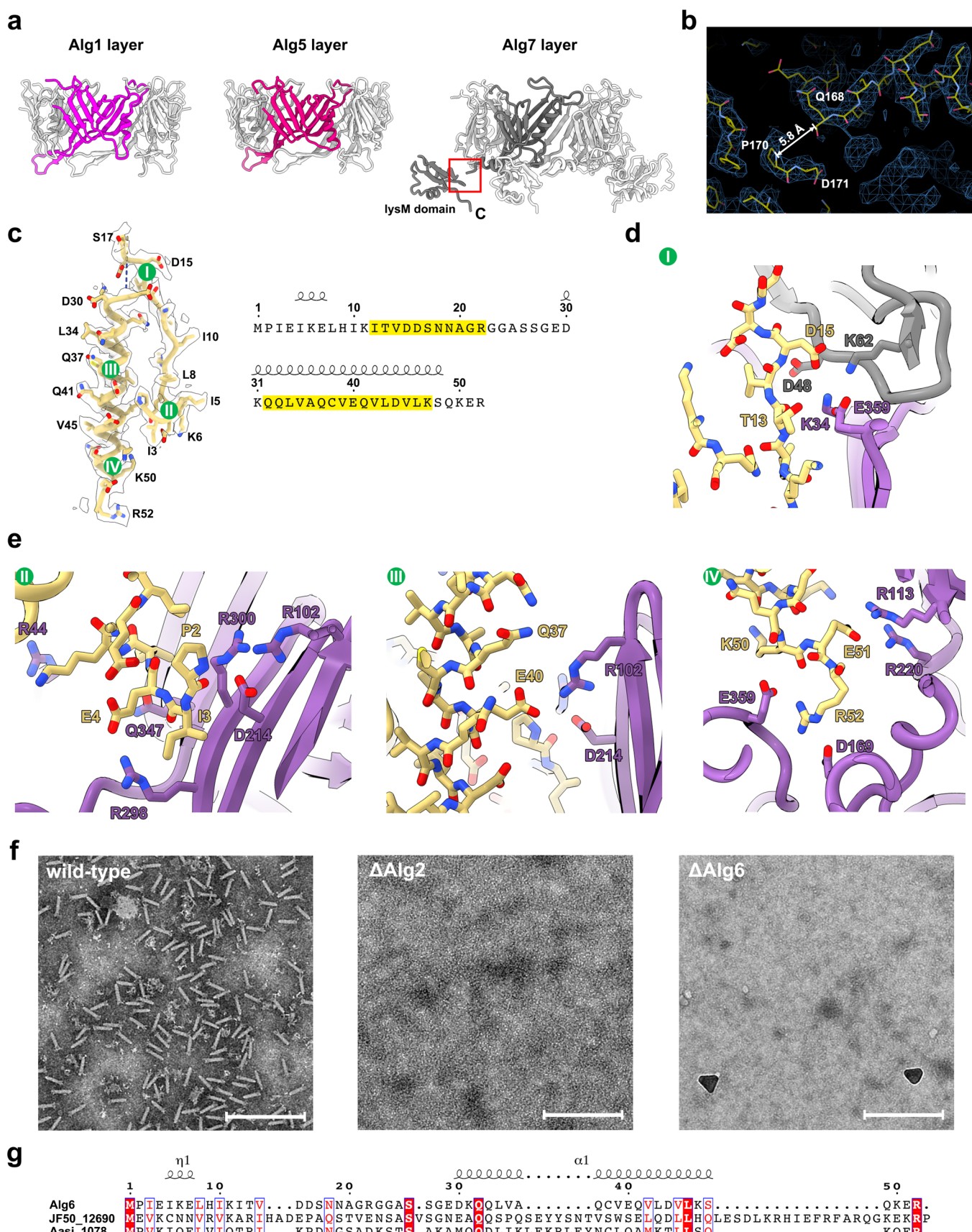

**a** Alg1 layer | Alg5 layer | Alg7 layer

lysM domain | C

**b** Q168 | P170 | 5.8 Å | D171

**c** S17 | D15 | I | D30 | L34 | I10 | Q37 | III | L8 | Q41 | II | I5 | V45 | I3 | IV | K6 | K50 | R52

1 ... 10 ... 20 ... 30
M P I E I K E L H I K ITVDDSNNAGR G G A S S G E D

31 ... 40 ... 50
K QQLVAQCVEQVLDVLK S Q K E R

**d** I | D15 | K62 | D48 | E359 | T13 | K34

**e** II | R44 | P2 | R300 | R102 | E4 | Q347 | I3 | D214 | R298

III | Q37 | E40 | R102 | D214

IV | R113 | K50 | E51 | R220 | E359 | R52 | D169

**f** wild-type | ΔAlg2 | ΔAlg6

**g**
η1 ... α1

1 ... 10 ... 20 ... 30 ... 40 ... 50
Alg6       MPIEIKELHIKITV...DDSNNAGRGGAS.SGEDKQQLVA.......QCVEQVLDVLKS..................QKER.
JF50_12690 MEVKCNNVRVKARIHADEPAQSTVENSASVSGNEAQQSPQSEYYSNTVSWSELQDLLHQLESDLKRHIEFRFARQGKEKRP
Aasi_1078  MPVKIQELVIQTRI...KPDNGSADKSTSAKAMQQDLIKLEKRLEYNCLQAMKTLLSQ..................KQER.

**Extended Data Fig. 6 | See next page for caption.**

**Extended Data Fig. 6 | Structures of the components of the tube initiator complex and the interactions between plug protein and other baseplate components. a:** Ribbon diagrams showing the structures of inner tube layer (Alg1) and two layers in tube initiator complex (Alg5 and Alg7). The color code for one subunit in each layer matches Fig. 1d. Red box indicates the position of the cleaved loop in Alg7, which is shown in panel **b**. **b:** Zoom-in of the loop in (**a**) showing that the distance between two residues (Q168 and P170) is too large to accommodate one amino acid residue (S169), implying that there is a potential cleaved site at this loop. **c:** Shadowed surface and ribbon diagram showing the structure of plug protein (Alg6), while the corresponding secondary structure and MS analyses is shown in the right panel. The residues are labeled and shown with side chains, with the disordered loop represented by a dashed line. The peptide fragments detected in MS are highlighted yellow and the secondary structural information is shown above the corresponding sequences based on the model. **d:** Stick and ribbon diagram showing the interactions between Alg6 (yellow), Alg7 (grey), and Alg8 (purple), which is corresponding to the part I in (**c**). Alg6 is shown in stick style, while other proteins are shown in ribbon style. The key residues mediating contacts are labeled and shown with side chains. **e:** Stick and ribbon diagrams showing the interactions between Alg6 (yellow) and Alg8 (purple), which is corresponding to the part II-IV in (**c**). Alg6 is shown in stick style and Alg8 is shown in ribbon style. The key residues mediating contacts are labeled and shown with side chains. **f:** Negative staining EM images of crude sheath preparation from *A. machipongonensis* wild-type and the related mutants showing that Alg6 is crucial for particle assembly. No particles were observed in the AlgoCIS ΔAlg6 mutant, which is similar to the AlgoCIS ΔAlg2 mutant. Bars: 500 nm. This experiment was performed three independent times. **g:** Sequence alignment of Alg6 and putative homolog proteins in MACs (JF50_12690) from *P. luteoviolacaea* and T6SS[iv] from *Ca*. Amoebophilus asiaticus (Aasi_1078). Identical residues are shown in white on a red background, while similar residues are shown in red. The blue boxes indicate the conserved positions. The secondary structure of Alg6 is shown above the corresponding sequences. The image is made from Esprit[78].

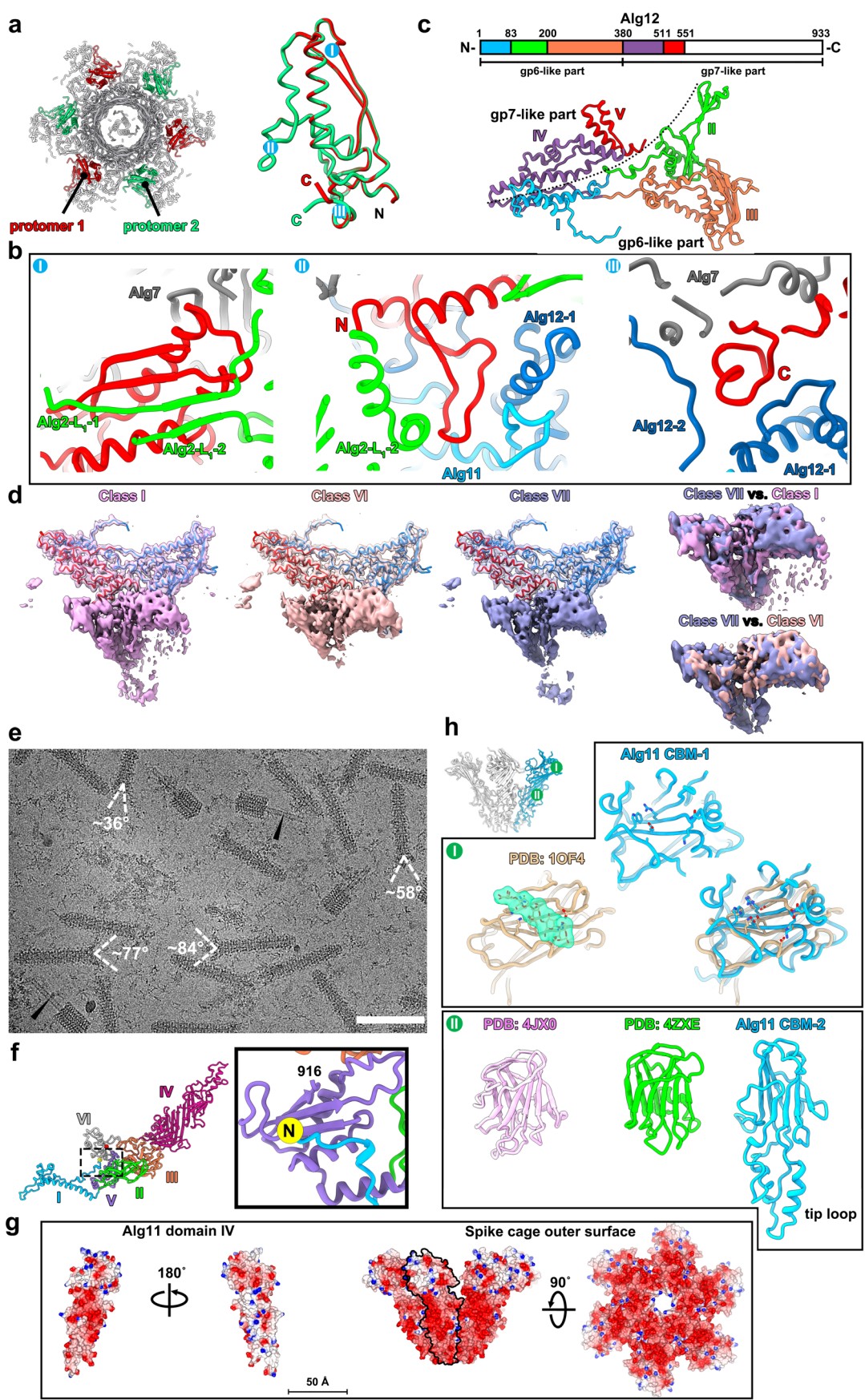

**Extended Data Fig. 7 | See next page for caption.**

**Extended Data Fig. 7 | Structural analyses of the AlgoCIS baseplate wedge. a:** Ribbon diagram of a perpendicular slice of the AlgoCIS baseplate showing the positions of two different Alg9 protomers (protomer 1: red; protomer 2: green). The other components are colored white. The structural superposition of two Alg9 protomers is shown in the right panel. **b:** Magnified views of ribbon diagrams showing the interactions between Alg9 (protomer1: red) and other baseplate components (Alg2-L$_1$, Alg7, Alg11, and Alg12), which are related to the parts I-III in (**a**). The color code for different proteins matches Fig. 1d. **c:** Schematic and ribbon diagrams showing the domain organization of Alg12. The C-terminal part without structural modeling is colored white. The boundary between gp6-like and gp7-like part is marked with dashed line. **d:** Shadowed surface and ribbon diagrams showing the structural dynamics of the peripheral baseplate in 3D classification analyses. The densities around built-up models are shown transparent, while the dynamic peripheral parts are shown in solid surface. The pair comparisons of dynamic parts between different classes are shown in the right panel. The gp6-like and gp7-like part of Alg12 model is colored blue and red. The color code for different classes matches the 3D classification part in Extended Data Fig. 2a. **e:** Representative cryoEM image showing the dynamics of tail fibers in the extended AlgoCIS particles. The angles in individual particles were measured and labeled. The lumen of expelled inner tubes (highlighted by black arrowheads) are observed empty upon firing. Bar: 50 nm. **f:** Ribbon diagram showing the domain organization of Alg11. The color code for different domains matches Fig. 4b. Box indicates the β-sheet assembly of the N-terminal strand and the domain V, which is shown in the right panel. **g:** Surface electrostatic potentials of the Alg11 domain IV and the overall outer surface of the spike cage revealing that the outer surface of the spike cage is dominated by negative charges, with a positively charged residue (Lys 572) located at the tip. The boundary of one protomer is marked by black contour. Negative and positive electrostatic potentials are colored red and blue, respectively. Bar: 50 Å. **h:** Ribbon diagrams showing the structures of carbohydrate binding modules (CBM1/2) in Alg11 and the related homologous proteins. Top-left: the overall structure of spike cage. Box I corresponding to part I in the top-left panel: ribbon diagrams showing that the CBM1 might potentially bind to sugar as the homologous protein (PDB entry: 1OF4)[39]. The surface of sugar is shown in green transparent. The structural superimposition reveals that the residues, participating in the contacts with substrates, are well-conserved in the CBM1. Box II corresponding to part II in the top-left panel: ribbon diagrams showing the structures of CBM2 and the homologous proteins (PDB entries: 4ZXE and 4JX0)[79].

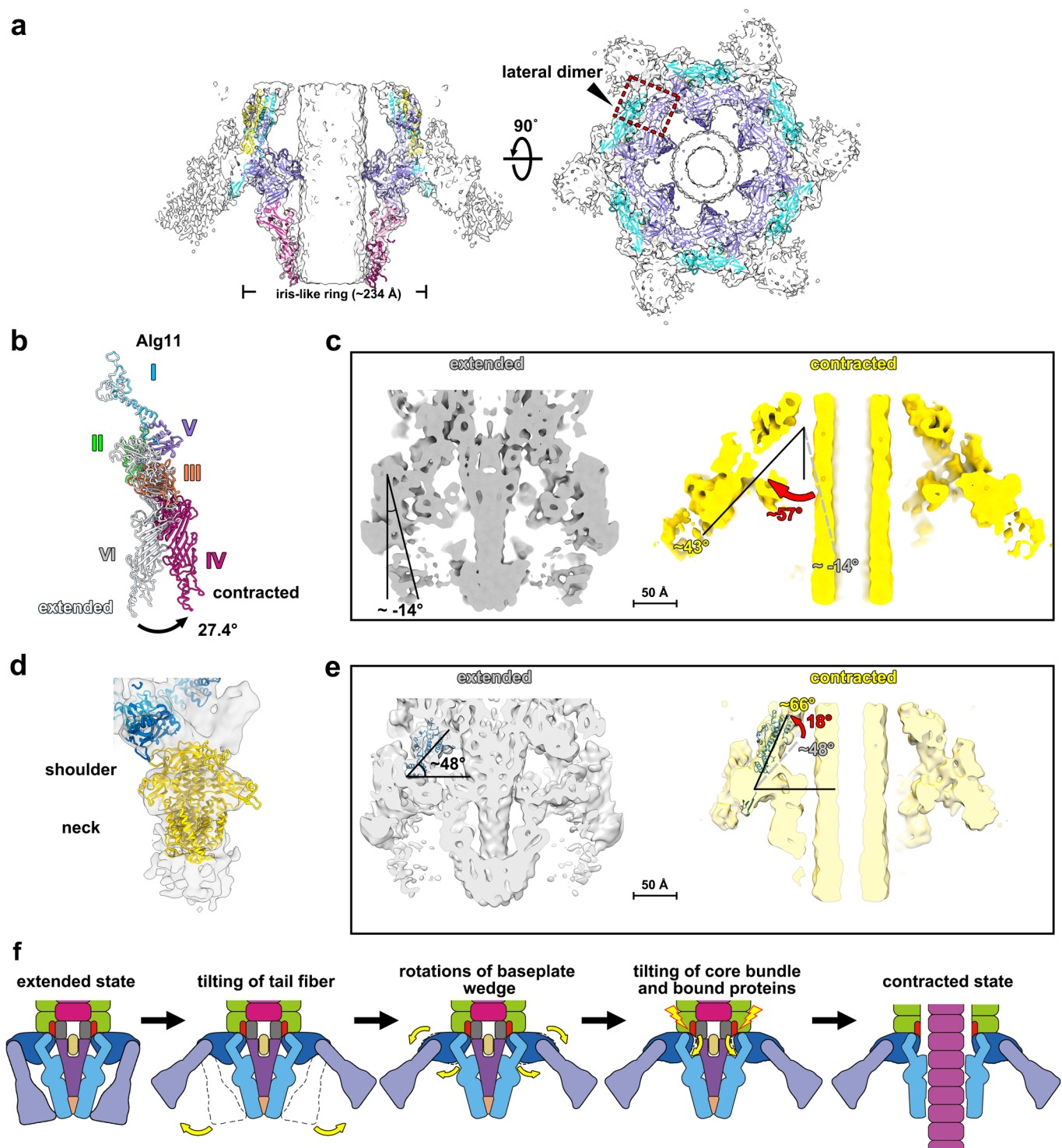

**Extended Data Fig. 8 | Conformational changes upon firing reveal the fate of the AlgoCIS cage. a:** Side view (left) and bottom view (right) of shadowed surface and ribbon diagrams, showing that the spike cage remains attached to the baseplate and opens. The iris-like ring is intact in contracted AlgoCIS. The color code matches Fig. 4a. The red dashed box indicates the position of one lateral dimer. **b:** Structural superpositions of Alg11 proteins in the extended and contracted states showing that the conformational change of domain IV mainly contributes to the opening of spike cage. The color code for different domains in the contracted state matches Fig. 4b, while the structure of Alg11 in the extended state is colored white. **c:** Cutaway view of shadowed surface diagrams of extended (grey) and contracted (yellow) AlgoCIS structures showing that the tail fiber has a large outward tilt upon firing. The extended and contracted maps were both lowpass-filtered to 10 Å. Bar: 50 Å. **d:** Shadowed surface and ribbon diagram showing that the docking of the T6SS TssK structure [gold, PDB entry: 5MWN[42]] fits the overall structure of the AlgoCIS tail fiber. The tail fiber has a similar shape as the shoulder domain in TssK. **e:** Cutaway views of shadowed surface and ribbon diagrams of extended (grey) and contracted (yellow) AlgoCIS structures showing that the baseplate core bundles have a higher tilt angle related to the plane of the iris-like ring after contraction. The extended and contracted maps were both lowpass-filtered to 10 Å. Bar: 50 Å. **f:** Schematic showing the putative contraction and signal transmission process in AlgoCIS upon firing. The color code matches Fig. 1d. The lightning bolts represent the conformational changes of the gp25-like protein Alg9, which triggers sheath contraction.

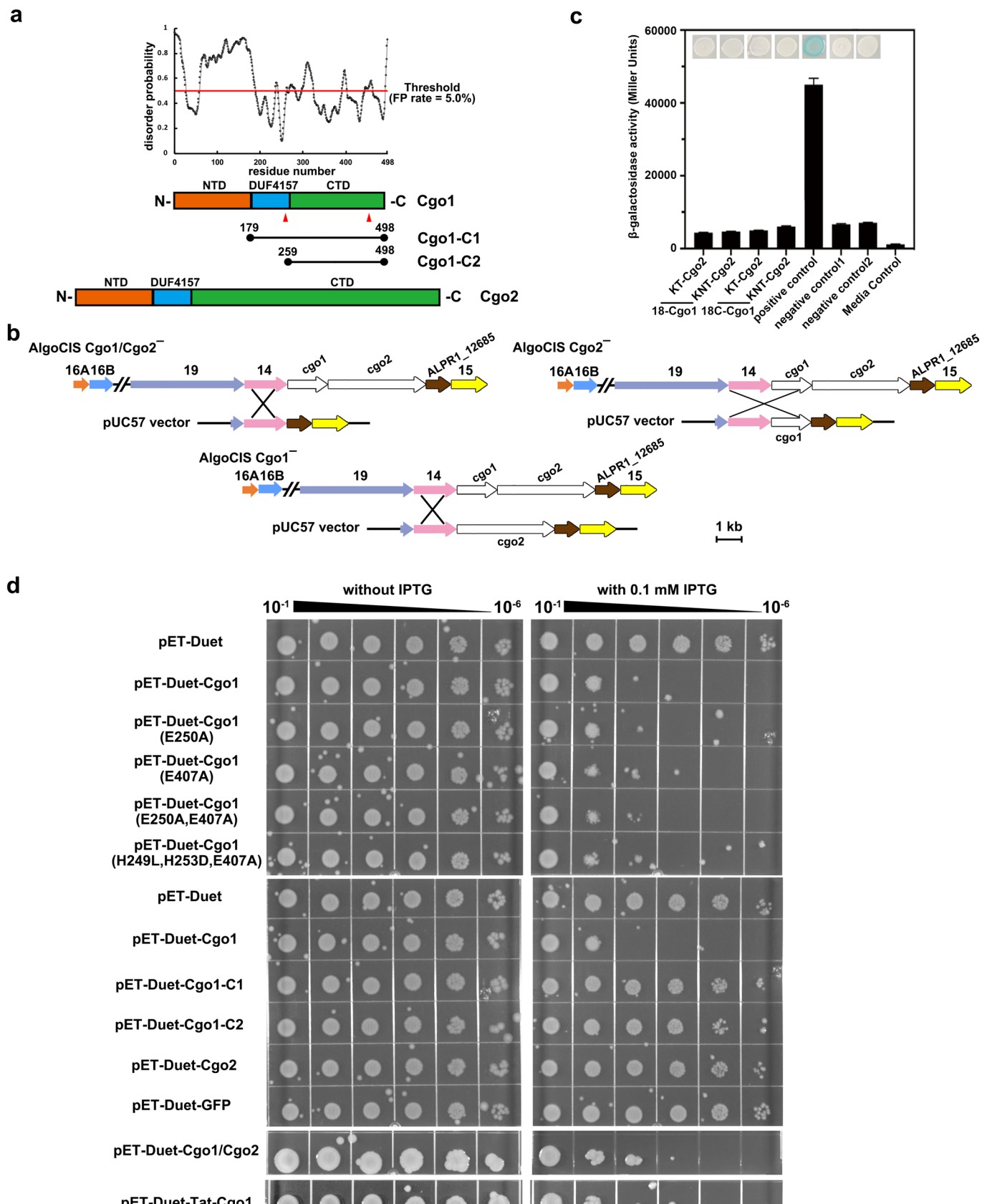

**Extended Data Fig. 9 | See next page for caption.**

**Extended Data Fig. 9 | Characterization and two-hybrid analyses of putative cargo proteins. a:** Schematic and disorder probability plot showing the domain organizations and disorder properties of putative cargo protein Cgo1 (ALPR1_12695). The positions of two putative metalloprotease motifs (HExxH) are highlighted by red triangles. The domain organizations of the other putative cargo protein Cgo2 (ALPR1_12690) is shown in the bottom panel. **b:** Schematic showing the principles of the cross-over methods to generate different cargo protein-deficient mutants. The color code for different genes matches Fig. 1a. **c:** Two hybrid analyses showing no interaction between the two cargo proteins. Shown are qualitative spot assay on a X-gal plate (top) and the quantification by using a ONPG β-galactosidase assay with fusions to N (KT/18 C) and C-terminal (KNT/18) adenylate cyclase subunits (bottom). Representative experiments are shown. Bars are mean with error bars indicating standard deviation. **d:** Bacterial spot assays show that the expression of full-length Cgo1 had a killing effect in *E. coli* (the columns represent the serial dilutions from $10^{-1}$ to $10^{-6}$ with a dilution factor 10 per step). Furthermore, the point mutants in putative metalloprotease motifs, or the co-expression with Cgo2, or the fusion with periplasmic tag did not impair the killing effects. The strains carrying pET-Duet/pET-Duet-GFP vectors are negative controls.

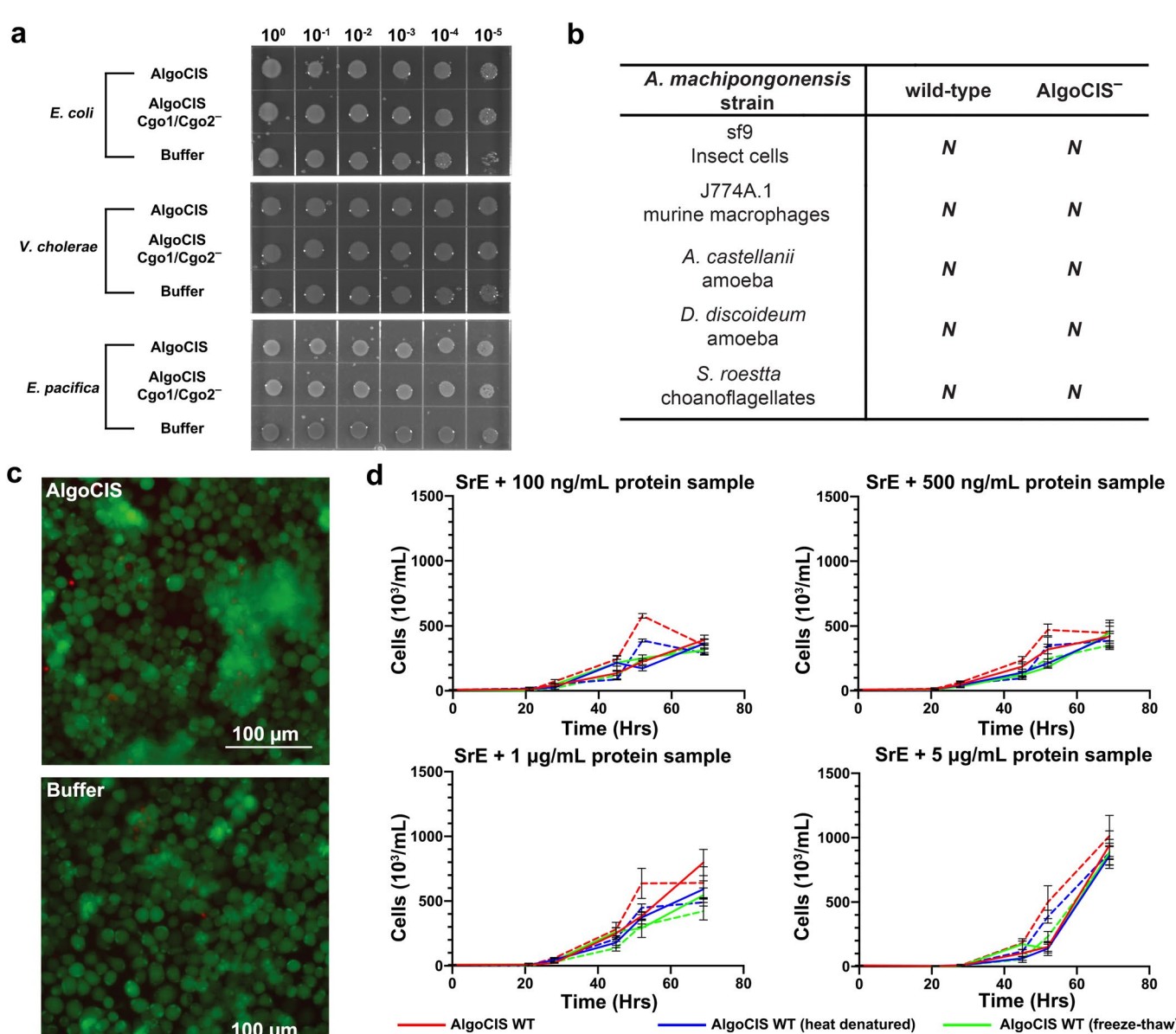

**Extended Data Fig. 10 | Co-incubation of different types of cells with the purified AlgoCIS or A. machipongonensis bacteria. a:** Bacterial killing assays showing that different bacterial strains (*E. coli*, *V. cholera* and *E. pacifica*) are insensitive to the treatment with AlgoCIS. The purified Cgo1/Cgo2⁻ mutant and buffer are regarded as negative controls. **b:** Table summarizing different eukaryotic cells that were subjected to the screen against *A. machipongonenesis* wild-type and AlgoCIS⁻ mutant. There was no significant difference in toxicity or lethal effects for all types of cells treated with wild-type/AlgoCIS⁻ bacteria (indicated by 'N'). **c:** Insect cell killing assay showing that the insect cells were not sensitive to the treatment with AlgoCIS. The buffer is regarded as negative control. Red (Propidium iodide): dead cells. Green (fluorescence diacetate): live cells. This experiment was performed three independent times with representative images shown. Bars: 100 μm. **d:** Growth curves of the choanoflagellate *S. rosetta* treated with purified AlgoCIS wild-type and Cgo1/Cgo2⁻ showing that there was no significant effect on the cellular proliferation of the choanoflagellate. The heat denaturation and freeze-and-thaw samples are regarded as negative controls. For the 5 μg/mL treatment, two biologically independent sample timepoints (n=2) were measured with technical replicates (n=3) and for lower concentration treatments (1 μg/mL, 500 ng/mL, 100 ng/mL), one biological sample was measured with technical replicates (n=3). Data are presented as mean values +/- SD of all technical replicates.

