## [Peer Review File. · Nature Microbiology]

Peer Review Information

Journal: Nature Microbiology

Manuscript Title: Identification and structure of an extracellular contractile injection system from the marine bacterium *Algoriphagus machipongonensis*

Corresponding author name(s): Martin Pilhofer

Editorial Notes:

Redactions – transferred manuscripts (mention of the other journal) This manuscript has been previously reviewed at another journal. This document only contains reviewer comments, rebuttal and decision letters for versions considered at Nature **XX**. Mentions of the other journal have been redacted.

Reviewer Comments & Decisions:

Decision Letter, initial version:

Dear Martin,

Thank you very much for your enquiry about submitting your manuscripts "Structure of a thylakoid-anchored contractile injection system" and "Identification and structure of an extracellular contractile injection system in marine bacteria" to Nature Microbiology. I've now had the chance to discuss the manuscripts, the reviews from {redacted} and your proposed rebuttal/revision plans with my colleagues, and we all agree that the papers are both really interesting and likely to appeal to our readers, so we would be happy to consider them for publication here. We also appreciate that generating additional insights into the biological functions of these systems may not be easy, but any data that you can add (as you already propose to do in the rebuttal) would certainly help to further strengthen the paper. Once you're ready to submit the revised papers, please use the link below and make sure to include a point-by-point rebuttal to all of the previous reviewer concerns, as we will try to use this information to speed up the editorial process.

In order to submit your complete manuscripts to Nature Microbiology, please use the link below:

{redacted}

In the meantime, if you have any questions, please feel free to contact me.

With best regards,

{redacted}

Decision Letter, first revision:

Dear Martin,

Thank you for submitting your revised manuscript "Identification and structure of an extracellular contractile injection system in marine bacteria" (NMICROBIOL-21092428A). It has now been seen by our referees and their comments are below. The reviewers find that the paper has improved in revision, and therefore we'll be happy in principle to publish it in Nature Microbiology, pending some minor revisions to satisfy the referees' final requests and to comply with our editorial and formatting guidelines.

2We will now perform detailed checks on your paper and all associated files, and will send you a checklist detailing our editorial and formatting requirements in about a week. Since the current version of your manuscript is in a PDF format, please email us a copy of the file in an editable format (such as Microsoft Word or LaTeX) so that we can perform these checks-- we can not proceed with PDFs at this stage. Please do not upload the final materials and make any revisions until you receive this checklist from us.

Thank you again for your interest in Nature Microbiology, and please do not hesitate to contact me if you have any questions.

{redacted}

Reviewer #1 (Remarks to the Author):

This study from Xu et al. provides a detailed structural examination of the contractile injection system (CIS) from the marine bacterium *Algoriphagus machipongonensis*. This is the first high-resolution structure of a CIS from clade Ib. It reveals several novel features and provides new insights into CIS evolution and specialization. As stated in our original review, the structural work is very high quality, and this study will be of high interest to researchers in the field. The revised manuscript has address all the issues we raised in our review, and thus, I strongly recommend publication of this paper in its current state.

Reviewed by Benjamin Engel
(Originally reviewed by Benjamin Engel and Ricardo Righetto)

Reviewer #2 (Remarks to the Author):

The paper covers some very fine structural work on a novel contractile injection system variant in a non-model marine bacterium. The main issue raised by previous reviewers was that the function of the novel CIS is unclear. That remains the case, but in general the authors look to have made a good response to the previous reviewers' comments, such that the revised manuscript does not contain unjustified statements about the function. The authors deserve great credit for devising experiments to test some possibilities for physiological function, in addition to all the structural work, the molecular phylogeny and the mutagenesis to probe the functions of specific elements of the novel CIS. It's a huge amount of work, and it's clear that the cryo-EM structural work has been done to a very high standard.

My only specific comments are on the assay for intoxication of *E. coli* by expression of the cargo protein (Fig. 5c; lines 355-362; response to a point from previous Reviewer #2). Firstly, the "N-

2terminal periplasmic tag" used is not properly explained anywhere, as far as I can see. Judging only from the axis label in Fig. 5c, it is a TAT leader sequence, but no detail is provided anywhere. The TAT system is easily saturated, meaning that for highly-expressed proteins it is often the case that only a small proportion ends up in the periplasm. Here there is no data on how effectively the cargo protein may have been exported to the periplasm, and therefore it is not possible to draw conclusions about relative toxicity in the periplasm vs the cytoplasm. The idea of the reviewer's suggestion was to see if Cgo1 would become more convincingly toxic when exported to the periplasm. In fact this seems to be a negative result, since the periplasmic translocation tag merely fails to mitigate the toxic effect (lines 360-362). Therefore I don't think the new experiment strengthens the authors' argument, and I agree with the previous reviewer #2 that this experiment does not provide clear evidence of antibacterial activity of Cgo1. There is no data on the level of the protein in the cells, or if the levels of GFP and Cgo2 are at all equivalent to the level of Cgo1 (they could be very different even if expressed off the same promoter and induced the same way). The killing effect looks modest, and could just be a non-specific effect of overexpression, as the previous reviewer says and as the authors acknowledge in their response. They say that they they have "toned down" their statements about the antibacterial effect of Cgo1, however there is still a bold headline saying that "expression of Cgo1 is toxic against bacteria" (line 354) and the assay occupies a big section of Fig. 5 in the main text. I suggest moving Fig. 5c and the detail of the intoxication assay to Supplementary and putting just a short statement in the main to text to point to the data and to explain that the killing and intoxication assays were inconclusive.

Reviewer #3 (Remarks to the Author):

The following notes and remarks cover both papers from the Pilhofer lab - by Xu et al. and by Weiss et al. - that described the structure of PVC-like complexes of Algoriphagus and Anabaena, respectively.

11
SEP

Despite not having access to the original version of the MSs, it is clear that the authors addressed the comments and critiques of the referees very thoroughly and improved their papers to a point where some minor edits can still be suggested, but they might be superfluous.

Obviously, the function of the two biological systems in question (two different phage tail-like structures) remains unknown. So, this is a major sticking point for both MSs. The description of the structure and illustrations are excellent.

To this CIS expert, the only major problem with both MSs is their Abstracts. The authors have long been proponents of sub-typing the class of contractile injection systems into smaller groups. With these two new papers and new systems (that are nevertheless are very much PCV-like), the authors emphasize in the abstracts how these new systems fit into that classification ("compatible" or "incompatible" with T6SS or eCIS). I do not think this is relevant because the classification is built using a very small dataset and hence can be incorrect. Both abstracts must emphasize what have been found in these papers rather than (dis)agreement with some classification system. The latter would have been acceptable for, and count even constitute the main subject of, a bioinformatic paper. But in this case, the Abstract must be a summary of experimental discoveries.

3Some nitpicking.

In the MS "Identification and structure of an extracellular contractile injection system in marine bacteria"

Line 49. Why is Ref. 9 cited? What the process of membrane puncturing studied in that MS?

Line 53. Not just "pyocins" (the term came out of nowhere), but R-type pyocins.

Line 220. This paper <https://www.nature.com/articles/415553a> (PMID: 11823865) is the most appropriate reference here.

Decision Letter, final checks:

Dear Martin,

Thank you for your patience as we've prepared the guidelines for final submission of your Nature Microbiology manuscript, "Identification and structure of an extracellular contractile injection system in marine bacteria" (NMICROBIOL-21092428A). Please carefully follow the step-by-step instructions provided in the attached file, and add a response in each row of the table to indicate the changes that you have made. Ensuring that each point is addressed will help to ensure that your revised manuscript can be swiftly handed over to our production team.

In recognition of the time and expertise our reviewers provide to Nature Microbiology's editorial process, we would like to formally acknowledge their contribution to the external peer review of your manuscript entitled "Identification and structure of an extracellular contractile injection system in marine bacteria". For those reviewers who give their assent, we will be publishing their names alongside the published article.

Nature Microbiology offers a Transparent Peer Review option for new original research manuscripts submitted after December 1st, 2019. As part of this initiative, we encourage our authors to support increased transparency into the peer review process by agreeing to have the reviewer comments, author rebuttal letters, and editorial decision letters published as a Supplementary item. When you submit your final files please clearly state in your cover letter whether or not you would like to participate in this initiative. Please note that failure to state your preference will result in delays in accepting your manuscript for publication.

4Cover suggestions

As you prepare your final files we encourage you to consider whether you have any images or illustrations that may be appropriate for use on the cover of Nature Microbiology. Covers should be both aesthetically appealing and scientifically relevant, and should be supplied at the best quality available. Due to the prominence of these images, we do not generally select images featuring faces, children, text, graphs, schematic drawings, or collages on our covers. We accept TIFF, JPEG, PNG or PSD file formats (a layered PSD file would be ideal), and the image should be at least 300ppi resolution (preferably 600-1200 ppi), in CMYK colour mode. If your image is selected, we may also use it on the journal website as a banner image, and may need to make artistic alterations to fit our journal style. Please submit your suggestions, clearly labeled, along with your final files. We'll be in touch if more information is needed.

Nature Microbiology has now transitioned to a unified Rights Collection system which will allow our Author Services team to quickly and easily collect the rights and permissions required to publish your work. Approximately 10 days after your paper is formally accepted, you will receive an email in providing you with a link to complete the grant of rights. If your paper is eligible for Open Access, our Author Services team will also be in touch regarding any additional information that may be required to arrange payment for your article. Please note that you will not receive your proofs until the publishing agreement has been received through our system.

Please note that Nature Microbiology is a Transformative Journal (TJ). Authors may publish their research with us through the traditional subscription access route or make their paper immediately open access through payment of an article-processing charge (APC). Authors will not be required to make a final decision about access to their article until it has been accepted. Find out more about Transformative Journals

Authors may need to take specific actions to achieve compliance with funder and institutional open access mandates. For submissions from January 2021, if your research is supported by a funder that requires immediate open access (e.g. according to Plan S principles) then you should select the gold OA route, and we will direct you to the compliant route where possible. For authors selecting the subscription publication route our standard licensing terms will need to be accepted, including our self-archiving policies. Those standard licensing terms will supersede any other terms that the author or any third party may assert apply to any version of the manuscript.

For information regarding our different publishing models please see our Transformative Journals page. If you have any questions about costs, Open Access requirements, or our legal forms, please contact ASJournals@springernature.com.

Please use the following link for uploading all the required materials:

{redacted}

With best regards,

{redacted}

Author Rebuttal, second revision:

Reviewers' comments in Black.

Authors' responses in Blue.

Reviewers' Comments:

Reviewer #1: This study from Xu et al. provides a detailed structural examination of the contractile injection system (CIS) from the marine bacterium *Algoriphagus machipongonensis*. This is the first high-resolution structure of a CIS from clade Ib. It reveals several novel features and provides new insights into CIS evolution and specialization. As stated in our original review, the structural work is very high quality, and this study will be of high interest to researchers in the field. The revised manuscript has address all the issues we raised in our review, and thus, I strongly recommend publication of this paper in its current state.

Reviewed by Benjamin Engel
(Originally reviewed by Benjamin Engel and Ricardo Righetto)

Thank you for the encouraging evaluation of our manuscript. We really appreciate your comments throughout peer review, which strengthened our paper significantly.

Reviewer #2: The paper covers some very fine structural work on a novel contractile injection system variant in a non-model marine bacterium. The main issue raised by previous reviewers was that the function of the novel CIS is unclear. That remains the case, but in general the authors look to have made a good response to the previous reviewers' comments, such that the revised manuscript does not contain unjustified statements about the function. The authors deserve great credit for devising experiments to test some possibilities for physiological function, in addition to all the structural work, the molecular phylogeny and the mutagenesis to probe the functions of specific elements of the novel CIS. It's a huge amount of work, and it's clear that the cryo-EM structural work has been done to a very high standard.

Thank you very much for taking the time to review our paper and sharing your supportive comments with us.

6My only specific comments are on the assay for intoxication of *E. coli* by expression of the cargo protein (Fig. 5c; lines 355-362; response to a point from previous Reviewer #2). Firstly, the "N-terminal periplasmic tag" used is not properly explained anywhere, as far as I can see. Judging only from the axis label in Fig. 5c, it is a TAT leader sequence, but no detail is provided anywhere. The TAT system is easily saturated, meaning that for highly-expressed proteins it is often the case that only a small proportion ends up in the periplasm. Here there is no data on how effectively the cargo protein may have been exported to the periplasm, and therefore it is not possible to draw conclusions about relative toxicity in the periplasm vs the cytoplasm. The idea of the reviewer's suggestion was to see if Cgo1 would become more convincingly toxic when exported to the periplasm. In fact this seems to be a negative result, since the periplasmic translocation tag merely fails to mitigate the toxic effect (lines 360-362). Therefore I don't think the new experiment strengthens the authors' argument, and I agree with the previous reviewer #2 that this experiment does not provide clear evidence of antibacterial activity of Cgo1. There is no data on the level of the protein in the cells, or if the levels of GFP and Cgo2 are at all equivalent to the level of Cgo1 (they could be very different even if expressed off the same promoter and induced the same way). The killing effect looks modest, and could just be a non-specific effect of overexpression, as the previous reviewer says and as the authors acknowledge in their response. They say that they they have "toned down" their statements about the antibacterial effect of Cgo1, however there is still a bold headline saying that "expression of Cgo1 is toxic against bacteria" (line 354) and the assay occupies a big section of Fig. 5 in the main text. I suggest moving Fig. 5c and the detail of the intoxication assay to Supplementary and putting just a short statement in the main text to point to the data and to explain that the killing and intoxication assays were inconclusive.

We revised our manuscript and moved the results of our spot assays to the Extended Data Figure 9d. In addition, we toned down our statement and only described the killing results of Cgo1 protein when expressed in *E. coli* (line 321-330).

Reviewer #3: The following notes and remarks cover both papers from the Pilhofer lab - by Xu et al. and by Weiss et al. - that described the structure of PVC-like complexes of Algoriphagus and Anabaena, respectively. 
Despite not having access to the original version of the MSs, it is clear that the authors addressed the comments and critiques of the referees very thoroughly and improved their papers to a point where some minor edits can still be suggested, but they might be superfluous.

Obviously, the function of the two biological systems in question (two different phage tail-like structures) remains unknown. So, this is a major sticking point for both MSs. The description of the structure and illustrations are excellent.

To this CIS expert, the only major problem with both MSs is their Abstracts. The authors have long been proponents of sub-typing the class of contractile injection systems into smaller groups. With these two new papers and new systems (that are nevertheless are very much PCV-like), the authors emphasize in the abstracts how these new systems fit into that classification ("compatible" or "incompatible" with T6SS or eCIS). I do not think this is relevant because the classification is built using a very small dataset and hence can be incorrect. Both abstracts must emphasize what have been found in these papers rather than (dis)agreement with some classification system. The latter would have been acceptable for, and could even constitute the main subject of, a bioinformatic paper. But in this case, the Abstract must be a summary of experimental discoveries.

Thank you very much for taking the time to review our paper. The mechanistic classification of CISs into eCISs and T6SSs is widely used. In our view, it is important to state whether the results obtained here are consistent or

inconsistent with the respective systems. Lines 26-29 clearly describe the detailed findings of this paper. We updated the abstract to reflect the editorial suggestions for changes.

Some nitpicking.

In the MS "Identification and structure of an extracellular contractile injection system in marine bacteria"

Line 49. Why is Ref. 9 cited? What the process of membrane puncturing studied in that MS?

Line 53. Not just "pyocins" (the term came out of nowhere), but R-type pyocins.

Line 220. This paper <https://www.nature.com/articles/415553a> (PMID: 11823865) is the most appropriate reference here.

Thank you for the comments. We addressed all points accordingly in the new version of the manuscript.

Final Decision Letter:

Dear Martin,

I hope you had a nice break and wish you a happy new year.

Sorry for the delay in getting back to you on this paper (our offices were closed over the last few weeks), but I am writing to say that we are very pleased to accept your Article "Identification and structure of an extracellular contractile injection system from the marine bacterium *Algoriphagus machipongonensis*" for publication in Nature Microbiology. Thank you for having chosen to submit your work to us and many congratulations to you and your co-authors.

After the grant of rights is completed, you will receive a link to your electronic proof via email with a request to make any corrections within 48 hours. If, when you receive your proof, you cannot meet this deadline, please inform us at rjsproduction@springernature.com immediately. You will not receive

8your proofs until the publishing agreement has been received through our system

Acceptance of your manuscript is conditional on all authors' agreement with our publication policies (see <https://www.nature.com/nmicrobiol/editorial-policies>). In particular your manuscript must not be published elsewhere and there must be no announcement of the work to any media outlet until the publication date (the day on which it is uploaded onto our website).

Please note that *Nature Microbiology* is a Transformative Journal (TJ). Authors may publish their research with us through the traditional subscription access route or make their paper immediately open access through payment of an article-processing charge (APC). Authors will not be required to make a final decision about access to their article until it has been accepted. [Find out more about Transformative Journals](https://www.springernature.com/gp/open-research/transformative-journals)

Authors may need to take specific actions to achieve compliance with funder and institutional open access mandates. For submissions from January 2021, if your research is supported by a funder that requires immediate open access (e.g. according to [Plan S principles](https://www.springernature.com/gp/open-research/plan-s-compliance)) then you should select the gold OA route, and we will direct you to the compliant route where possible. For authors selecting the subscription publication route our standard licensing terms will need to be accepted, including our [self-archiving policies](https://www.springernature.com/gp/open-research/policies/journal-policies). Those standard licensing terms will supersede any other terms that the author or any third party may assert apply to any version of the manuscript.

We welcome the submission of potential cover material (including a short caption of around 40 words) related to your manuscript; suggestions should be sent to Nature Microbiology as electronic files (the image should be 300 dpi at 210 x 297 mm in either TIFF or JPEG format). Please note that such pictures should be selected more for their aesthetic appeal than for their scientific content, and that

9colour images work better than black and white or grayscale images. Please do not try to design a cover with the Nature Microbiology logo etc., and please do not submit composites of images related to your work. I am sure you will understand that we cannot make any promise as to whether any of your suggestions might be selected for the cover of the journal.

Congratulations once again to you and your co-authors on putting together such a nice story, I look forward to seeing it published.